# A Comprehensive Assessment of Tropical Stratospheric Upwelling in Specified Dynamics CESM1.2.2 (WACCM)

Nicholas A. Davis[1,a], Sean M. Davis[2], Robert W. Portmann[2], Eric Ray[1], Karen H. Rosenlof[2], and Pengfei Yu[1,b]

[1]University of Colorado Cooperative Institute for Research in Environmental Sciences (CIRES) at the

NOAA Earth System Research Laboratory (ESRL) Chemical Sciences Division, Boulder, CO, USA

[2]NOAA Earth System Research Laboratory (ESRL) Chemical Sciences Division, Boulder, CO, USA

[a]*Current affiliation*: Atmospheric Chemistry Observations and Modeling Laboratory, National Center for Atmospheric Research, Boulder, CO, USA

[b]*Current affiliation*: Institute for Environmental and Climate Research, Jinan University, Guangzhou, GD, China

*Correspondence to*: Nicholas A. Davis (nadavis@ncar.edu)

**Abstract.** Specified dynamics (SD) schemes relax the circulation in climate models toward a reference meteorology to simulate historical variability. These simulations are widely used to isolate the dynamical contributions to variability and trends in trace gas species. However, it is not clear if trends in the stratospheric overturning circulation are properly reproduced by SD schemes. This study assesses numerous SD schemes and modeling choices in the Community Earth System Model (CESM) Whole Atmosphere Community Climate Model (WACCM) to determine a set of best practices for reproducing interannual variability and trends in tropical stratospheric upwelling estimated by reanalyses. Nudging toward the reanalysis meteorology as is typically done in SD simulations does not accurately reproduce lower stratospheric upwelling trends present in the underlying reanalysis. In contrast, nudging to anomalies from the climatological winds or anomalies from the zonal mean winds and temperatures better reproduces trends in lower stratospheric upwelling, possibly because these schemes do not disrupt WACCM's climatology. None of the schemes substantially alter the structure of upwelling trends - instead, they make the trends more or less AMIP-like. An SD scheme's performance in simulating the acceleration of the shallow branch of the mean meridional circulation from 1980-2017 hinges on its ability to simulate the downward shift of subtropical lower stratospheric wave momentum forcing. Key to this is *not* nudging the zonal-mean temperature field. Gravity wave momentum forcing, which drives a substantial fraction of the upwelling in WACCM, cannot be constrained by nudging and presents an upper-limit on the performance of these schemes.

## 1 Introduction

Stratospheric ozone loss has been halted by a concerted international effort to eliminate emissions of ozone-depleting substances under the Montreal Protocol [WMO, 2018]. While ozone is recovering in the upper stratosphere, there is some indication of a decline in the tropical lower stratosphere since the late 1990's when ozone depleting substance emissions peaked [Ball et al. 2018]. Ozone in this region is strongly mediated by the vertical advection of ozone-scarce

tropospheric air by the residual circulation - the wave-driven, thermally-indirect overturning circulation of the stratosphere [Butchart 2014, and references therein] - as well as eddy mixing of ozone-rich air from the extratropical lower stratosphere [Abalos et al. 2013].

Ball et al. [2018] evaluated two specified dynamics (SD) simulations to assess this unexpected decline. In SD simulations, a climate model's circulation is nudged toward the meteorology of an atmospheric reanalysis. The goal of such simulations is to constrain the known variability of the atmospheric circulation to better isolate the role of chemical processes, insofar as the reanalysis meteorology is reliable. Ball et al. [2018] found that their SD simulations were unable to reproduce an observed decline in lower stratospheric ozone, and offered several explanations, including emissions of short-lived ozone depleting substances, the quality of the meteorological data, and the quality of model-simulated tracer transport in the lower stratosphere. However, other studies using a chemical transport model [Chipperfield et al. 2018] and an SD-like model simulation [Wargan et al. 2018] found that observed meteorological variability could explain the recent changes in stratospheric ozone, with some disagreement on the sign of the change in the tropical lower stratosphere.

By construction nudging schemes target each model's prognostic variables, which are generally either the horizontal winds or vorticity and divergence, often in addition to temperature, surface pressure, and occasionally specific humidity. Both Van Aalst et al. [2004] and Loffler et al. [2016] nudged surface pressure and vorticity, which characterize the balanced flow, stronger than they nudged divergence and temperature. The motivation for such a scheme is that the essence of dynamical variability is the evolution of the large-scale balanced flow, and that model physics like convection should govern shorter time-scale variability in the thermodynamics and the unbalanced flow. Schmidt et al. [2018] nudged only the horizontal winds in CESM to assess the freely-evolving temperature response to volcanic aerosols, while Solomon et al. [2015] and [2016] nudged horizontal winds and temperatures to assess polar ozone and heterogeneous chlorine chemistry in the TTL, both of which require an accurate reproduction of absolute temperature. But from any sampling of past studies, it is difficult to understand the impact of the peculiarities of each nudging scheme on simulated dynamics, transport, and chemistry.

Lower stratospheric ozone is sensitive to modes of natural variability including the El Niño-Southern Oscillation [Randel et al. 2009, Diallo et al. 2018], which can drive variations in tropical stratospheric upwelling via vertical shifts in gravity wave momentum forcing [Calvo et al. 2010]. Regardless of the interpretation of recent ozone variability, the question remains, "why is there a discrepancy between the Ball et al. [2018] simulations and the simulations in Chipperfield et al. [2018] and Wargan et al. [2018]?" One possibility is that one or more of these schemes may not be reproducing the mean meridional circulation trends in the input meteorological data, which would present a critical flaw to their purported ability to constrain circulation variability. There is emerging evidence that nudging tends to increase the inter-model spread in measures of the residual circulation and chemical transport [Orbe et al. 2017, Orbe et al. 2018, Chrysanthou et al. 2019]. This is because the schemes do not perfectly reproduce dynamical variability, with substantial errors in the residual circulation [Akiyoshi et al. 2016]. Further, nudging - which is often implemented as a relaxation term - is substantially less sophisticated than 3-dimensional data assimilation, which itself tends to degrade model performance in simulating the stratospheric age of air [Meijer et al. 2004].

Chemical transport models, which do not prognostically model the atmosphere but instead directly ingest the meteorology, are not as susceptible to these errors [Chipperfield et al. 1999, Mahieu et al. 2014]. However, they must still assess vertical motion either through mass continuity or diabatic heating, in addition to estimating convective transport [Stockwell and Chipperfield 1999, Chipperfield 2005]. It is also possible that reanalyses contain spurious circulation trends that cannot (and perhaps should not) be reproduced by climate models [Chemke and Polvani 2019], given the range of trends in the residual circulation [Abalos et al. 2015; Kobayashi and Iwasaki 2016].

Here we pose a simple question: do nudging schemes reproduce the variability and trends in the mean meridional circulation of the input reanalysis meteorology? If not, what can be done to improve that representation? Given that multi-decadal trends in the earth system tend to be the residual of a balance of much larger terms, we hypothesize that disagreements between the climatologies of the input meteorology and the nudged model may lead to spurious circulations that interfere with upwelling trends. In this study, we examine one of the models and nudging schemes employed in Ball et al. [2018], the Community Earth System Model-Whole Atmosphere Community Climate Model (CESM (WACCM)), analyzing a series of nudging experiments to assess the impact of various modeling choices and nudging scheme variations on tropical stratospheric upwelling trends. We find that *not* nudging zonal mean temperatures results in the best reproduction of upwelling trends, while nudging zonal mean temperatures tends to degrade these trends, consistent with our hypothesis.

## 2 Model

We use version 1.2.2 of CESM (WACCM) [Marsh et al. 2013] using the Community Atmosphere Model Version 4 (CAM4) [Neale et al. 2013] finite volume dynamical core [Lin 2004], covering an altitude range of 0 to 140 km on a 1.9x2.5 degree grid. All of our simulations are atmosphere-only experiments, with prescribed sea surface temperatures (SST's), Coupled Model Intercomparison Project Phase 5 (CMIP5) (Taylor et al. 2015) historical and RCP4.5 fixed lower boundary conditions, time-varying solar forcings, and volcanic aerosols. The default WACCM-SD scheme nudges horizontal winds and temperatures by 1% per timestep, which, for a timestep of 30 minutes, corresponds to a nudging timescale of 50 hours. We attempted to run WACCM at up to 10% per timestep (or, 5 hour timescale), but this required increasing the physics parameterization sub-cycling due to convective scheme errors - the "nsplit" parameter. Such simulations are not numerically comparable so we have chosen to avoid assessing the impact of nudging timescale, though it is known to have varied impacts [Merryfield et al. 2013, Hardiman et al. 2017, Orbe et al. 2017]. WACCM is nudged toward the MERRA2 reanalysis instantaneous assimilation ("ASM") product [Gelaro et al. 2011], with meteorological input supplied to the model every 3 hours on 72 hybrid levels. The default nudging is a simple relaxation term computed by linearly interpolating the meteorological input between meteorological times. Surface pressure is prescribed based on MERRA2 meteorology, but allowed to vary sufficiently to ensure mass conservation [Lamarque et al. 2012, and references therein]. MERRA2 surface geopotential, surface wind stress, surface latent and sensible heat fluxes, ice and ocean grid fractions, and surface skin temperature are also incorporated into the scheme to ensure consistency.

CAM4 WACCM has 66 native levels. However, standard practice is to run the SD simulations on reanalysis levels, the implicit motivation being that the reanalysis meteorology does not need to be interpolated before running the simulations

[Marsh 2011, Chandran et al. 2013, Verronen et al. 2016]. In this configuration, the model instead runs on 88 levels - 72 levels from the surface to the lower mesosphere, on MERRA2 hybrid levels, with a further 16 free-running levels in the upper atmosphere (Fig. S1 and S2). This substantially impacts cloud heating and gravity wave momentum forcing, in part because the MERRA2 grid triples the number of model levels in the boundary layer and lower troposphere (Fig. S3). In addition to the standard SD simulations with 88 levels, we test whether running the model on these non-native levels impacts the resolved circulation and its trends by nudging WACCM on its 66 native levels with interpolated MERRA2 meteorology (which interpolates MERRA2 meteorology from 72 to 46 levels, with the remaining 20 levels above MERRA2's lid). It is worth noting that while the number of levels in MERRA2 in the troposphere is much greater than in WACCM at its native resolution, the levels and their spacing correspond quite closely in the upper troposphere to the mid-stratosphere, such that any differences in the circulation in that region are likely to be driven from below.

Standard practice is to nudge toward the actual reanalysis meteorology, but this could create conflicts as WACCM and MERRA2 have different climatologies. As an example, the tropical lapse-rate tropopause is nearly 1 km higher in WACCM than in MERRA2, so standard nudging might disrupt WACCM's tropical tropopause height and cold-point temperature, which will go on to impact stratospheric water vapor, ozone chemistry, and transport. Such artificial disruptions to the climate will engage the feedbacks that stabilize WACCM around its free-running climate, potentially interfering with upwelling trends. Further, even modern reanalyses exhibit spurious global discontinuities in zonal-mean temperatures and winds due to satellite transitions, so it may be a valid choice to avoid nudging the zonal mean all together [Long et al. 2017].

As a solution, we test a scheme which nudges WACCM toward climatological anomalies, rather than the full meteorological reanalysis fields [Zhang et al. 2014]. Monthly-mean seasonal cycles of U, V, and T are calculated in WACCM and MERRA2 and interpolated to daily values using cubic splines. This ensures a smooth annual cycle, avoids sampling issues, and only requires monthly-mean output. The annual cycle is calculated from 1980-2017 inclusive, from MERRA2 and from WACCM in free-running mode with 66 native levels and prescribed historical sea surface temperatures, hereafter "WACCM AMIP" (AMIP referring to the Atmospheric Model Intercomparison Project [Gates 1992]). To generate the nudging input, 3-hourly MERRA2 U, V, and T anomalies are calculated by subtracting the MERRA2 climatology from the MERRA2 meteorology, and then adding this anomaly to the CESM climatology. Explicitly, for a given field X at some point in level-latitude-longitude ($\sigma, \phi, \lambda$), the nudging tendency is given as

$$\frac{d}{dt}X_{nudge}(\sigma,\phi,\lambda) = \frac{1}{\tau}(X_{WACCM}(\sigma,\phi,\lambda) - (X_{MERRA2}(\sigma,\phi,\lambda) - \overline{X_{MERRA2}(\sigma,\phi,\lambda)} + \overline{X_{AMIP}(\sigma,\phi,\lambda)})) \quad \text{(Eq. 1)}$$

where the overbar indicates the climatological value of X and $\tau$ is the nudging timescale.

While this climatological anomaly nudging scheme in theory better preserves the climatology than nudging toward the actual reanalysis meteorology, it still nudges the zonal mean. As the stratospheric circulation is wave-driven, the most important aspect of the circulation that must be reproduced in order to reproduce trends in stratospheric upwelling are the resolved wave momentum forcings (gravity wave momentum forcing is parameterized and cannot be nudged). We therefore test an additional scheme wherein only the zonal anomalies are nudged in WACCM. This allows WACCM to freely model the zonal-mean circulation and climate, bypassing differences in the climatologies of the input meteorology and WACCM and limiting the influence of spurious reanalysis trends and features. As this scheme does not nudge the zonal-mean winds, it

could be combined with a separate QBO nudging scheme, and would also not disrupt a spontaneous QBO [Garcia and Richter 2019]. Explicitly, for a given field X at some point in level-latitude-longitude $(\sigma, \phi, \lambda)$, the nudging scheme is given as

$$\frac{d}{dt}X_{nudge}(\sigma, \phi, \lambda) = \frac{1}{\tau}((X_{WACCM}(\sigma, \phi, \lambda) - [X_{WACCM}](\sigma, \phi)) - (X_{MERRA2}(\sigma, \phi, \lambda) - [X_{MERRA2}](\sigma, \phi)))$$

(Eq. 2)

where the vertical brackets indicate the zonal mean value of X. An advantage of this method over climatological anomaly nudging is that it does not require substantial preprocessing and a reference AMIP simulation, but it does require source code modification as the zonal mean of a given field in the model and input reanalysis meteorology must be calculated on-line at every time step.

The final dimension along which we test SD simulations is the subset of nudged variables. While standard practice is to nudge U, V, and T, it is unclear which, if any, are most important for reproducing past variability and trends. To gain further insight, we tested several other combinations of nudging variables – UV, UT, and VT. We found that UT nudging does not constrain the meridional circulation (as might be expected), and found that VT nudging is too similar to UVT nudging to warrant further investigation (probably because the zonal-mean zonal winds are strongly constrained by temperature through geostrophy). For simplicity, we ignore these other combinations of variables and focus only on UVT and UV.

We also consider a simulation identical to the AMIP run, but with Quasi-biennial Oscillation (QBO) nudging in the tropics, designated "AMIPQBO". The free-running version of CESM 1.2.2 (WACCM) does not spontaneously generate a QBO, instead requiring a scheme to nudge the tropical zonal winds to the observed QBO.

Our full set of WACCM simulations is documented in Table 1. All simulations are run from January 1, 1980 to December 31, 2017.

| Short name | Case | Nudging variables | Vertical levels |
|---|---|---|---|
| AMIP | AMIP | None | 66 |
| AMIPQBO | AMIPQBO | U (tropics only) | 66 |
| UVT L88 | UVT L88 | U, V, T | 88 (MERRA2 levels) |
| UV L88 | UV L88 | U, V | 88 (MERRA2 levels) |
| UVT | UVT | U, V, T | 66 |
| UV | UV | U, V | 66 |
| $U_{ca}V_{ca}T_{ca}$ | UVT climatological anomaly | U, V, T climatological anomalies | 66 |

| $U_{ca}V_{ca}$ | UV climatological anomaly | U, V climatological anomalies | 66 |
| $U_{za}V_{za}T_{za}$ | UVT zonal anomaly | U, V, T zonal anomalies | 66 |
| $U_{za}V_{za}$ | UV zonal anomaly | U, V zonal anomalies | 66 |

Table 1: Full list of WACCM simulations examined in this study, the MERRA2 variables with which they are nudged, the number of vertical levels, and the short name by which the figures refer to the simulation. See text for further information.

## 3 Methods

The Transformed Eulerian Mean (TEM) residual circulation, an estimate of the mean meridional mass circulation, is  calculated by solving the zonal momentum and mass balance equations so that different physical contributions to trends in tropical upwelling can be parsed directly. The "downward control" principle (Haynes et al., 1991) relates the steady-state zonally averaged vertical mass transport at any level to the vertically-integrated wave forcing in the column above. If one relaxes the steady-state constraint, the residual circulation streamfunction can be diagnosed similarly from the sum of the time tendency of the zonal wind and the vertically-integrated wave forcing in the column above [Randel et al. 2002, Abalos et al. 2013], and is given by

$$\Psi*(p,\phi) = \frac{1}{g}\int_p^0 \left\{ \frac{a^2 cos^2(\phi)((a cos(\phi))^{-1}\nabla \cdot \vec{F} + [D_{gw}] - [u]_t)}{[m]_\phi} \right\}_{\phi=\phi(p')} dp'$$

(Eq. 3)

where $\Psi*$ is the TEM residual circulation streamfunction, $a$ is the radius of the earth, $p$ is the pressure, $\phi$ is latitude, $g$ is the acceleration due to gravity, $\vec{F}$ is the Eliassen-Palm (EP) flux vector, $D_{gw}$ is the subgrid-scale (gravity) wave momentum forcing, $u$ is the zonal wind, $m$ is the angular momentum per unit mass, and the subscripts outside of the brackets indicate derivatives. The vertical integral is computed along lines of constant angular momentum by interpolating all fields from latitude to angular momentum space, with $p'$ denoting the pressure element along isolines of angular momentum. All calculations are performed in pressure space to ensure consistency with WACCM's and MERRA2's vertical discretization using 12 month low-pass filtered monthly-mean fields, primarily to stabilize the integration along angular momentum contours. Eddy fluxes are calculated every 3 hours in MERRA2 on native levels, while eddy fluxes are output as a monthly-mean value in WACCM. We use averaged output for zonal-means and gravity wave tendencies. For illustrative purposes, we

display the approximate altitude, in addition to the pressure, by showing the average of the geopotential height from 30S to 30N in the AMIP simulation. Angular momentum per unit mass in the shallow atmosphere approximation is given by

$$m = a\cos(\phi)\left(u + \Omega a\cos(\phi)\right) \tag{Eq. 4}$$

where $\Omega$ is the rotation rate of the Earth. The meridional and vertical components of the Eliassen-Palm flux are given by the isobaric coordinate version of Equations 25-26 in Hardiman et al. [2010],

$$F^{(\phi)} = a\cos(\phi)\left([u]_p \frac{[v'\theta']}{[\theta]_p} - [u'v']\right) \tag{Eq. 5}$$

$$F^{(z)} = a\cos(\phi)\left(\left\{f - \frac{1}{a\cos(\phi)}\frac{\partial}{\partial\phi}\left([u]\cos(\phi)\right)\right\}\frac{[v'\theta']}{[\theta]_p} - [u'\omega']\right)$$

(Eq. 6)

where $f$ is the Coriolis parameter, $v$ and $\omega$ are the meridional wind and pressure velocity, and $\theta$ is the potential temperature. See Hardiman et al. [2010] for a description of the divergence operator in Eq. 3.

The total upwelling mass flux through the tropical lower stratosphere is a useful metric for the strength of the residual circulation as it diagnoses the total tropospheric air mass entering the stratosphere from below. This air is low in ozone, which is efficiently produced in the stratosphere [Fueglistaler et al. 2009], higher in chlorofluorocarbons and other chlorinated species, which can destroy polar ozone [Tegtmeier et al. 2016], and retains a record of the time the air entered the stratosphere based on the concentrations of gases with linearly-increasing tropospheric concentrations, such as sulfur-hexafluoride [Linz et al. 2017] (a correction for non-linearity is given by Garcia et al. [2011]), or based on gases with a strong seasonal cycle, such as water vapor [Mote et al. 1996]. The total upwelling mass flux, hereafter simply "upwelling", convolves the speed of the upward circulation with its spatial extent, and is expressed as in Rosenlof [1995] as

$$M^*(p) = 2\pi a(max[\Psi^*(p,\phi)] - min[\Psi^*(p,\phi)]) \tag{Eq. 7}$$

All trends are computed with linear least-squares fits. Statistical significance of trends is assessed with two-sided Student's t-tests using each sample's effective degrees of freedom estimated through its lag-1 autocorrelation, while a test for the difference of means is used to assess differences in the climatology.

## 4 Results

Upwelling monotonically decreases with altitude from a peak of $20 \times 10^9$ kg/s at 120 hPa (15.5 km) in the tropical tropopause layer (TTL) to $0.5 \times 10^9$ kg/s at 1 hPa (47.7 km) in the upper stratosphere, indicating divergent poleward flow at all levels (Fig. 1). The difference in upwelling between MERRA2 and AMIP is at a maximum in the TTL and approaches zero at 20 hPa, indicating that MERRA2 has less total mass outflow in the TTL and lower stratosphere than AMIP – a

weaker residual circulation shallow branch (the outflow is proportional to the derivative of the mass flux with respect to pressure).

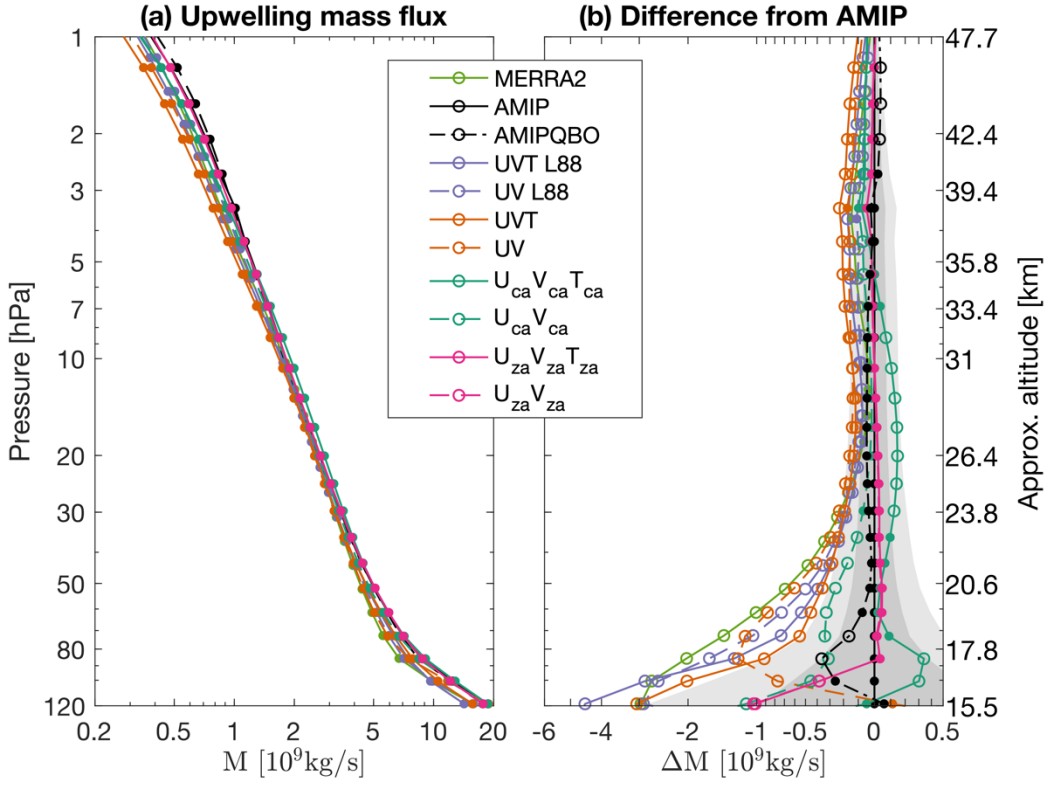

*Figure 1: Tropical upwelling mass flux (a) climatology and (b) difference from the AMIP simulation. Dark and light shading indicate 1- and 1.96-times the standard error. Open circles in (b) indicate the average is statistically significantly different from the AMIP simulation at the 95% confidence level.*

In the lower stratosphere, the upwelling in the standard nudging simulations (UV/T L88 and UV/T) falls between the weak upwelling in MERRA2 and the strong upwelling in the AMIP simulation, with the upwelling generally falling outside of the standard error of the AMIP simulation. By contrast, the upwelling in the climatological anomaly and zonal anomaly simulations is well within the standard error bounds, with the UV and UVT zonal anomaly nudging simulations most similar to AMIP over most of the stratosphere (the upwelling in the two zonal anomaly simulations is nearly identical). There are minor differences between the upwelling in AMIP and AMIPQBO, except in the TTL where AMIPQBO has weaker upwelling near the cold point. Across all nudging varieties, UVT nudging tends to have faster upwelling in the upper TTL and lower stratosphere than UV nudging. Overall, the runs which do not nudge the climatology generally preserve WACCM's free running climatological tropical upwelling mass flux.

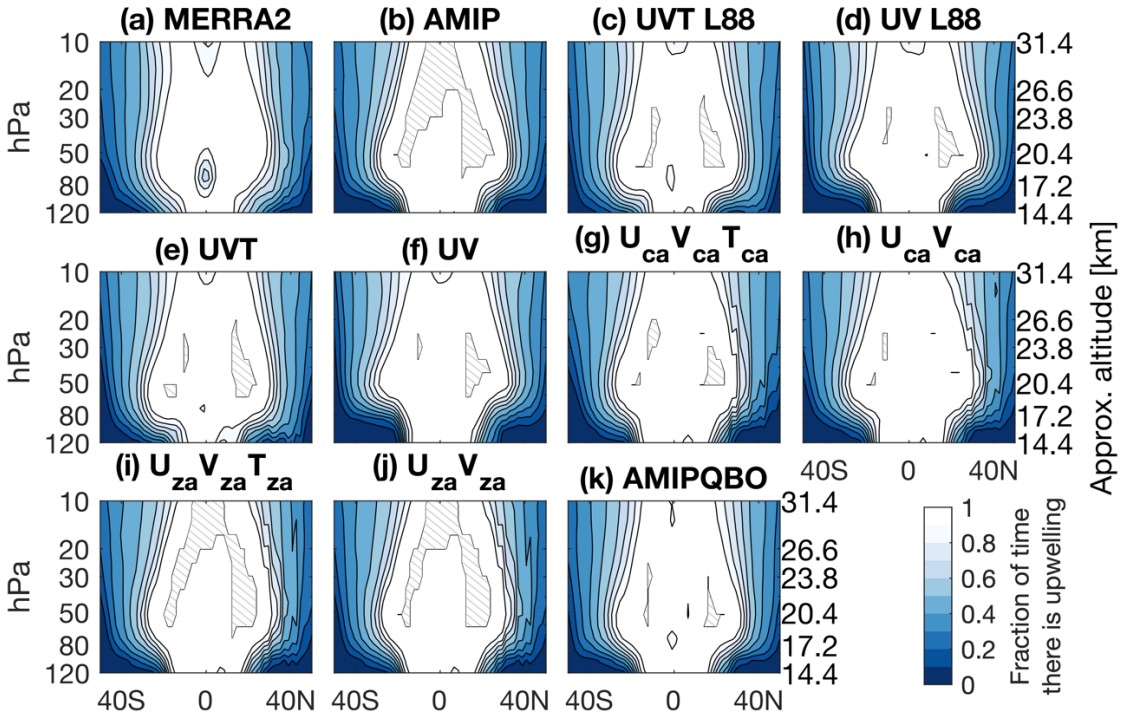

*Figure 2: Frequency of upwelling based on the streamfunction definition of the residual circulation for each model simulation. Hatching indicates there is upwelling 100% of the time.*

A statistical measure of tropical upwelling in the latitude-pressure plane is the frequency with which upwelling motion is observed (Fig. 2). Here, the TEM residual vertical velocity is calculated using the definition

$$[\omega]^* = [\omega] + \frac{1}{a\cos(\phi)} \left( \frac{[v'\theta']}{[\theta]_p} \cos(\phi) \right)$$

Major differences exist between WACCM and MERRA2. The upwelling frequency in MERRA2 tends to be split in the middle stratosphere, whereas there is consistent upwelling in both AMIP and AMIPQBO, indicating that the deep branch of the circulation in MERRA2 tends to be more consolidated in latitude in the winter seasons (see Fig.'s S4 and S5). While AMIP has constant upwelling (100% of the time) throughout the tropics, AMIPQBO only has scattered regions of constant upwelling in the subtropics. The secondary meridional circulations associated with the QBO are apparently strong enough to drive downwelling 1-5% of the time on the equator. In the standard nudging simulations, the morphology of upwelling appears as the average of the AMIP simulations and MERRA2, with slightly-split upwelling in the middle stratosphere and pockets of constant subtropical upwelling. The middle stratospheric upwelling in both of the anomaly nudging varieties appears more similar to AMIP than to MERRA2, further evidence that these nudging schemes preserve WACCM's climate. Upwelling in the zonal anomaly simulations is constant in the tropics as it is in AMIP, because neither have a QBO. While

the local minimum in upwelling in MERRA2 on the equator in the lower stratosphere may be related, in part, to the anomalous circulations associated with the QBO (compare AMIP and AMIPQBO), it is also possibly related to the minimum in ozone leading to a local minimum in diabatic heating [Ming et al. 2016].

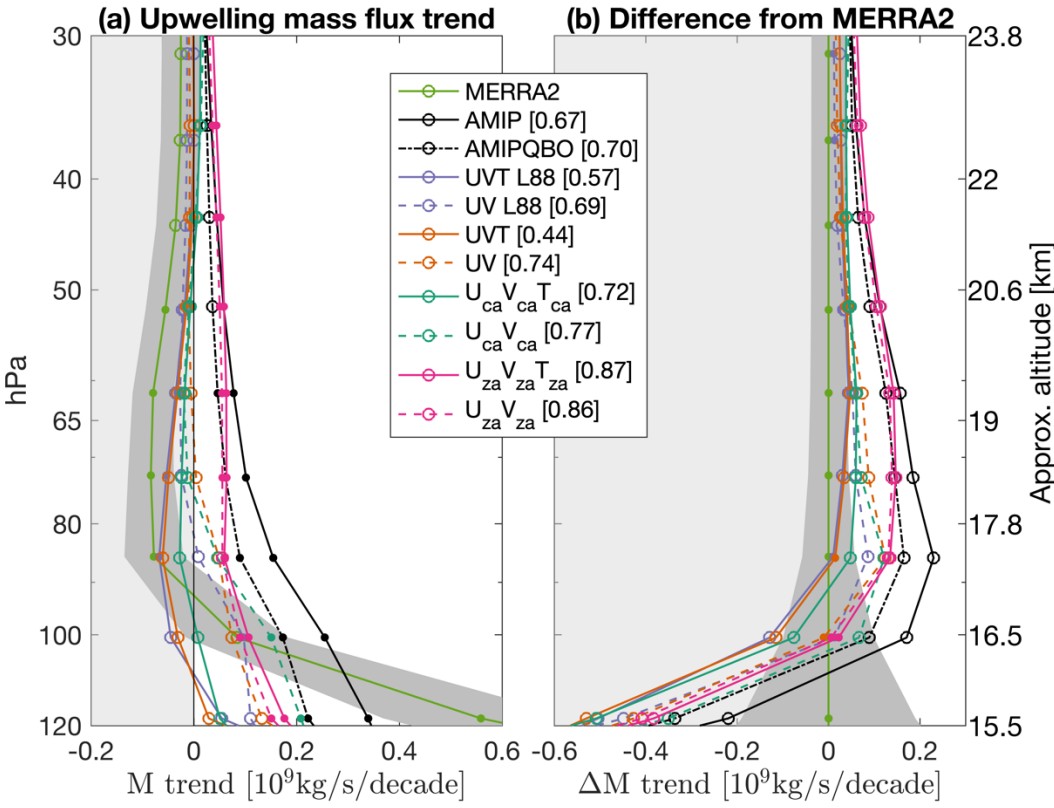

Figure 3: Upwelling mass flux (a) trends and (b) trend differences from MERRA2. Shading indicates the 95% confidence interval on the MERRA2 trends. Numbers in the legend indicate the correlation coefficient squared between each simulation's trends and the trends in MERRA2. Open circles in (a) indicate trends not statistically significant at the 95% confidence level, while open circles in (b) indicate trends statistically significantly different from MERRA2 at the 95% confidence level.

In similar fashion to the upwelling frequency, there are categorical differences between the upwelling trends in all WACCM simulations and MERRA2 over the full 1980-2017 analysis period (Fig. 3). MERRA2 shows a substantial upwelling acceleration in the TTL and upwelling deceleration in the lower and middle stratosphere. Physically, this pattern indicates enhanced tropospheric air mass transport into the TTL and a shift of upward mass transport from the deep to the shallow branch of the residual circulation. The upwelling trends in all WACCM runs tend to be smaller in the TTL and larger aloft compared to MERRA2, and there are no simulations that appear structurally different from each other. In fact, the

behavior of the upwelling trends in the WACCM simulations can be well-described by a simple linear shift in the trend pattern.

The UVT L88, UVT, and UVT climatological anomaly nudging simulations tend to reproduce the slightly negative-to-zero MERRA2 upwelling trend in the lower and middle stratosphere, while they completely fail to reproduce the acceleration in the TTL. Standard UVT nudging on both MERRA2 and native levels even suggests a decrease in the upwelling mass flux in the TTL, while the UVT climatological anomaly nudging suggests no significant trend. Upwelling trends in the standard UV and UV climatological anomaly nudging simulations is shifted to more positive values, with upwelling trends matching MERRA2 at 100 hPa, but underestimating the trend below and overestimating the trend aloft. There is no distinction between the upwelling trends in the UVT and UV zonal anomaly simulations, with both indicating acceleration at all levels. One objective measure of the disagreement in the morphology (and not sign) of the upwelling trends is the correlation coefficient squared ($R^2$) between the trends in the nudging simulations and the trends in MERRA2 (Fig. 3). In general, the UV nudging schemes show better agreement, especially for the standard nudging simulations, while the zonal anomaly nudging simulations show the best agreement of all (probably due to the kinked structure in the trend, with a minimum at 85 hPa similar to MERRA2). This all demonstrates that (incidentally) nudging zonal-mean MERRA2 temperatures - the UVT, UVT L66, and $U_{ca}V_{ca}T_{ca}$ simulations - has a negative impact on upwelling trend morphology and magnitude in the TTL. While it is true that the trends in the zonal anomaly nudging simulations are too positive above the TTL, key for constituent transport into the stratosphere and for recent ozone trends is the upwelling trend at and above the tropopause.

How can we understand the tendency for WACCM to produce such a different upwelling trend structure than MERRA2? In its free-running mode, with or without QBO nudging, WACCM produces an acceleration in upwelling throughout the TTL and the stratosphere (consistent with another AMIP simulation with different initial conditions; not shown), with a slightly stronger acceleration in the TTL. As nudging is introduced, first nudging only the resolved waves ($U_{za}V_{za}T_{za}$ and $U_{za}V_{za}$) and then only the zonal-mean and resolved wave horizontal winds (UV L88, UV, and $U_{ca}V_{ca}$), the trends become more negative (Fig. 4). The more comprehensive nudging, nudging both zonal-mean and resolved wave temperatures and horizontal winds (UVT L88, UVT, and $U_{ca}V_{ca}T_{ca}$), leads to even more negative trends. The only simulations with TTL and lower stratospheric trends unexplained by a regression of all WACCM simulations are the zonal anomaly nudging simulations, which fall outside of the prediction interval. In general, then, nudging schemes are unable to change the structure of the upwelling trends that are inherent to WACCM over this analysis period, only their average value. MERRA2's trend structure exists in a completely different phase space, well outside of the range of behavior predicted by the WACCM simulations. The upwelling trends in the UV nudging schemes tend to be "closest" to the trends in MERRA2, but they cannot be said to reproduce these trends.

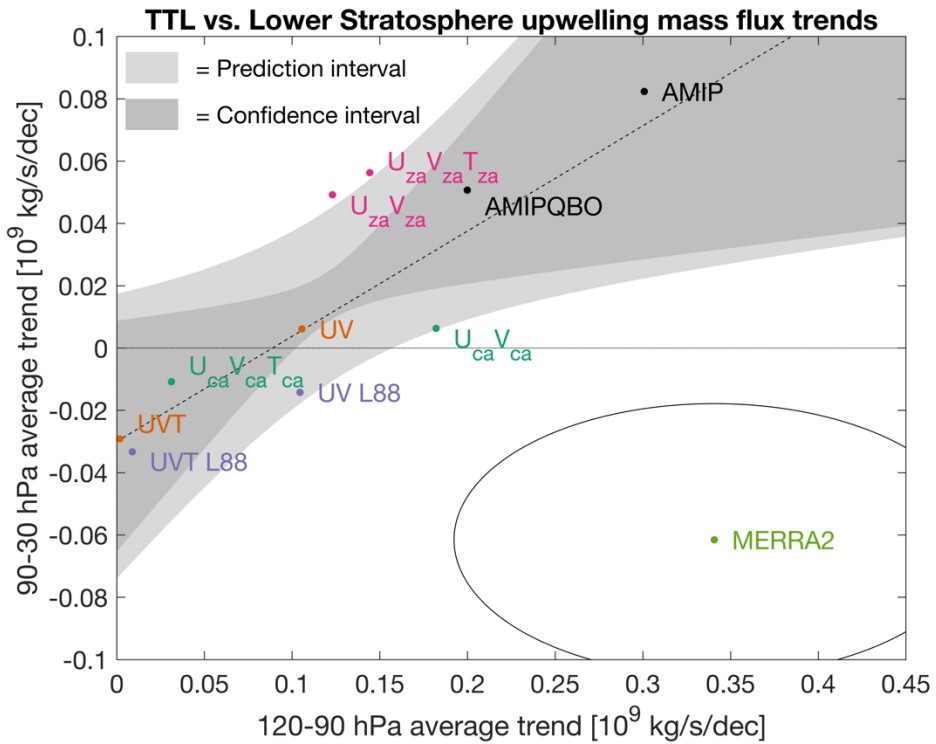

*Figure 4: Tropical upwelling trends averaged over the lower stratosphere versus upwelling trends averaged over the TTL. The regression line, 95% confidence interval, and prediction interval are indicated by the dashed line, dark shading, and light shading, respectively. The circle around MERRA2 indicates the 95% confidence interval.*

### 4.1 Variability

To gain some further insight, it is worth examining the time series of upwelling mass flux anomalies at 85 hPa at the top of the TTL. At this level, MERRA2 and the standard UVT nudging simulations indicate deceleration and the other simulations indicate acceleration. Upwelling variability in the standard nudging simulations, which comprehensively nudge the meteorology in both the time (climate and anomaly) and spatial (zonal mean and resolved wave) dimensions, agrees well with the variability in MERRA2 (Fig. 5). Anomalies in the UV nudging simulations tend to be more negative than those in the UVT nudging simulations and MERRA2 early in the record, and more positive later in the record, leading to a more positive trend in the UV nudging simulations at this altitude (Fig. 3).

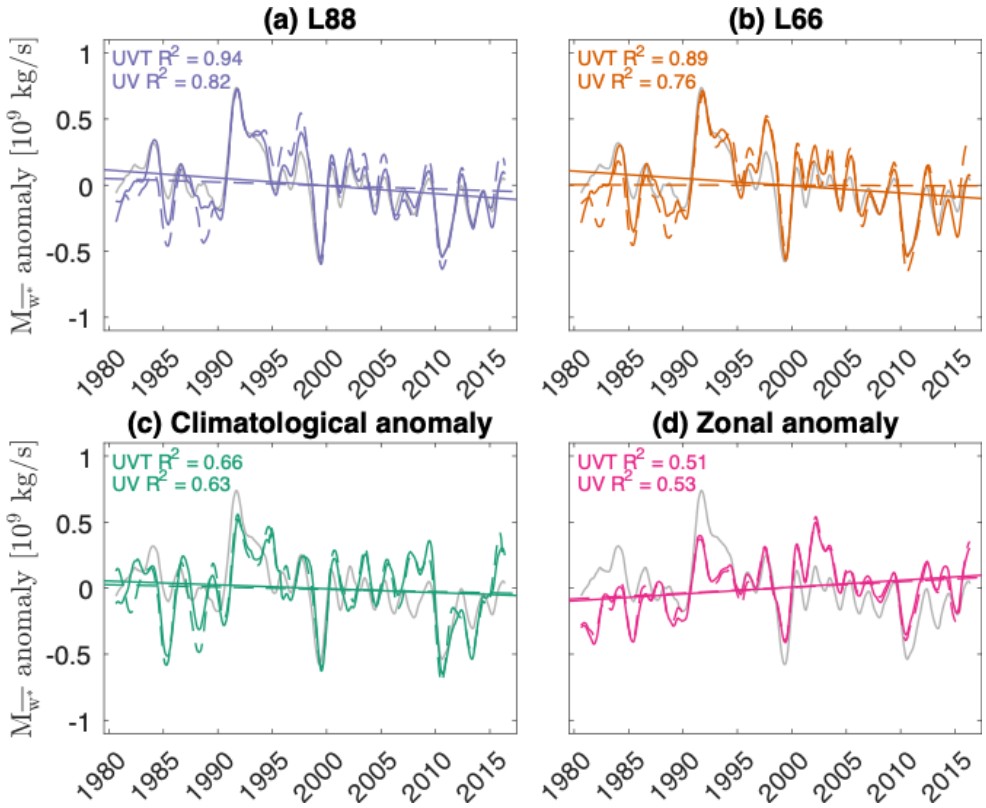

*Figure 5: 12-month low-pass filtered time series of tropical upwelling mass flux and its linear trend at 85 hPa in the (a) standard L88 nudging, (b) standard nudging, (c)* climatological *anomaly nudging, and (d) zonal anomaly nudging simulations, with UVT and UV nudging indicated by the solid and dashed lines and MERRA2 indicated by the solid grey line. The correlation coefficient squared between each time series and MERRA2 is shown in each panel.*

Climatological anomaly nudging constrains the temporal anomalies in both the zonal mean and the resolved waves. At first glance, then, it is surprising that the anomalies in the upwelling mass flux in the climatological anomaly nudging simulations disagree with those in MERRA2. In particular, while the magnitude of the anomalies seems consistent between the two, the peaks in the climatological anomaly simulations tend to be shifted relative to those in MERRA2. Similarly, while the magnitude of the anomalies in the zonal anomaly nudging simulations is much weaker than in MERRA2, the peaks are more aligned. This suggests that the zonal mean and its climatology, in particular, can substantially dictate the projection of zonal mean and resolved wave variability onto upwelling. This inference is obvious from the momentum balance equation - the vertical and meridional shears in the zonal-mean zonal wind and the static stability kinematically determine the projection of the eddy heating, and dynamically determine wave propagation by modulating potential vorticity gradients. In other words, the same resolved wave variability produces different upwelling for different climates. This raises the question of whether it is even possible to construct a nudging scheme that reproduces both variability and trends from a model with a different climatology.

Below 85 hPa, the zonal anomaly nudging exhibits extremely poor correlations with the variability in MERRA2 and the other simulations, suggesting a strong role for the zonal-mean circulation in transforming wave dynamics into zonal-mean momentum forcing and therefore upwelling (Fig. 6). Prescribing either the zonal-mean circulation or its climatological anomaly seems to guarantee better performance, suggesting that the TTL upwelling is governed by interannual variability in the zonal-mean circulation. Above 85 hPa, there is the opposite behavior - zonal anomaly nudging performs slightly worse than the conventional nudging and substantially better than the climatological anomaly nudging. How could it be that prescribing both the zonal-mean climate and its variability, or neither, is better than prescribing only the variability? A plausible hypothesis is that there may be physical incoherencies in the climatological anomaly nudging simulations, arising from the combination of the zonal-mean anomalous circulation in MERRA2 with the zonal-mean climatological circulation in WACCM.

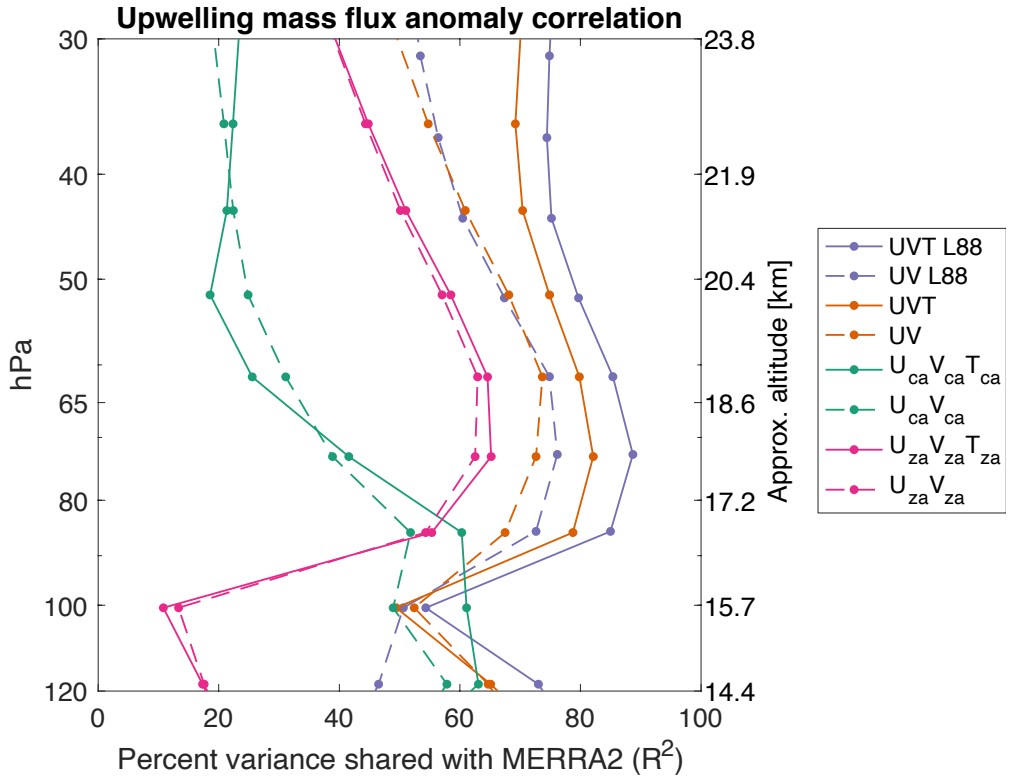

Figure 6: Correlation of tropical upwelling mass flux anomalies between each simulation and MERRA2.

## 4.2 Physical drivers of upwelling trends

Examining the contribution of specific terms in the momentum balance to the upwelling trends may provide some cursory answers as to why these nudging schemes produce disparate trends and variability. The upwelling trend due to a particular term of interest is given by the linearization about the total trend,

$$Trend_{term}(p) = Trend_{total}(p) - Trend_{detrendedterm}(p)$$

where $Trend_{total}(p)$ is the total upwelling trend, $Trend_{detrendedterm}(p)$ is the upwelling trend where the term of interest has been detrended, and $Trend_{term}(p)$ is the upwelling trend due to the term of interest. To detrend, we remove the linear trend in the term, including its intersect, and then add the mean of the trended data. This ensures that the term has the same average value before and after detrending.

First, we examine the contributions in the momentum balance equation (Fig. 7). Here, the contribution from [$u$] is with respect to its time tendency and the meridional angular momentum gradient - it does not consider the role of the shear terms in the Eliassen-Palm flux divergence. The total upwelling trend and its variation among the simulations is dominated by the contribution from resolved wave momentum forcing. However, gravity wave momentum forcing from orographic and non-orographic waves drives a surprisingly large fraction of the upwelling trend in the WACCM simulations - up to 100% of the trend above 70 hPa in the zonal anomaly simulations. In MERRA2, there is no detectable gravity wave contribution to the upwelling trend at any altitude, despite having an overhauled gravity wave scheme that minimizes analysis tendencies in the tropics [Molod et al. 2015, Coy et al. 2016]. This partly explains why the WACCM simulations tend to have a more positive upwelling trend than MERRA2 in the lower stratosphere and a more negative trend in the TTL.

Next, we examine the contributions to the upwelling trend from specific terms in the Eliassen-Palm flux divergence (Fig. 8). There is no contribution from the zonal-mean zonal wind, and while the vertical eddy momentum flux term drives deceleration in the TTL, its spread among the WACCM simulations and MERRA2 is small. There is some spread among the WACCM simulations due to the zonal-mean potential temperature term in the TTL, which curiously implicates different zonal-mean temperature trends. In general, though, it is the meridional eddy heat and momentum fluxes that drive the bulk of the variation among the WACCM simulations and between the WACCM simulations and MERRA2. The meridional eddy momentum flux drives an acceleration in upwelling in all of the WACCM simulations except UVT L88, but at a substantially weaker rate than in MERRA2 in the TTL. However, most of the spread among the WACCM simulations is due to the meridional eddy heat flux term, which drives deceleration in MERRA2 and the UVT nudging simulations and drives acceleration in all of the UV and the UVT zonal anomaly nudging simulations. This suggests that the zonal-mean temperature determines whether the eddy heat flux will drive acceleration or deceleration. It is not a kinematic impact (or else it would manifest in the upwelling trend due to zonal-mean potential temperature), but rather an impact on the projection of the wave physics within the simulations.

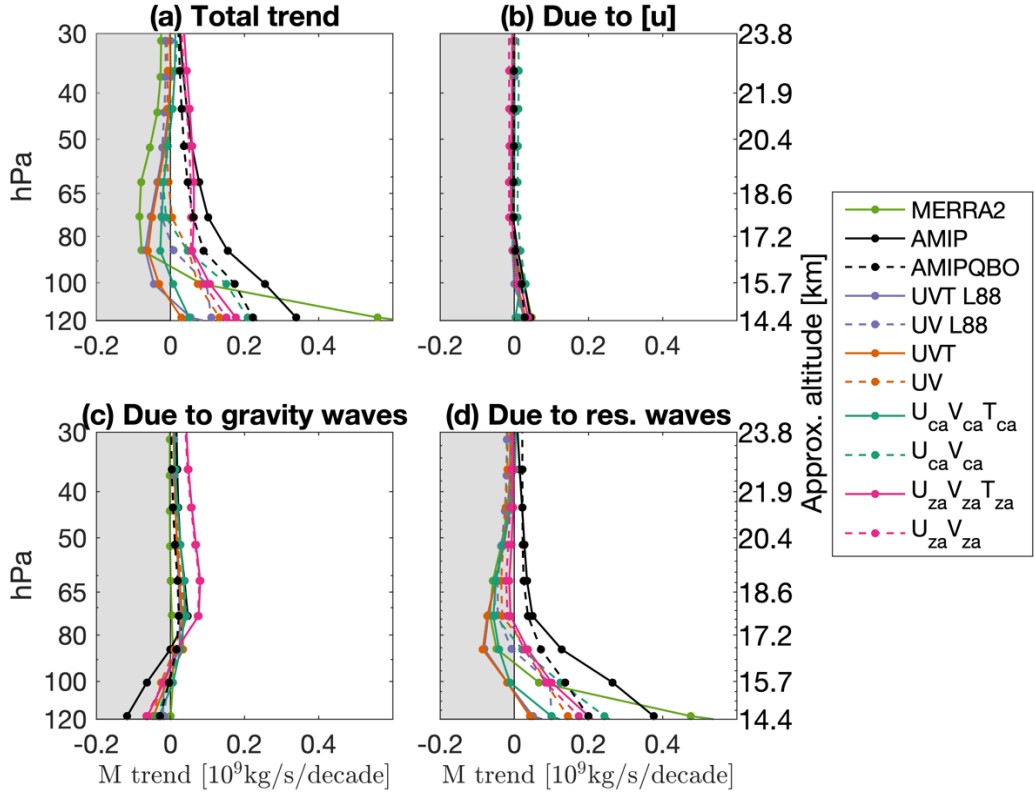

*Figure 7: Tropical upwelling mass flux trends: (a) total trend, (b) trend due to zonal-mean zonal wind, (c) trend due to gravity wave momentum forcing, and (d) trend due to resolved wave momentum forcing. See text for details.*

To gain a more comprehensive understanding, we examine the trends in the Eliassen-Palm flux and its divergence, as well as changes in the index of refraction (Fig.'s 9-10). The Eliassen-Palm flux is parallel to the group velocity for linear Rossby waves, such that Rossby wave generation, dissipation, and propagation can be assessed from the combined flux and divergence pattern [Edmon et. al 1980, and references therein]. These quantities directly diagnose the wave dynamics, while the index of refraction diagnoses whether the Rossby wave solution to the potential vorticity equation is oscillatory or evanescent. Waves will tend to refract toward positive values of the index, while negative values indicate that waves will evanesce exponentially away from their source region. Here, we take the difference of the quasi-geostrophic form of the index between 1994-2017 and 1980-1993 [Matsuno 1970]. Because of differencing the wavenumber term vanishes and the resulting changes in the index are applicable to all wavenumbers, though we note the upwelling in the TTL is primarily driven by planetary-scale waves [Ortland and Alexander 2014, Kim et al. 2016]. We also plot the latitudes of the streamfunction maxima and minima, indicative of where the mean meridional circulation vertical velocity switches from upward to downward, commonly called the "turn-around latitudes" [Rosenlof 1995], as the total upwelling mass flux within the pipe can be diagnosed solely by the net momentum tendency along this contour.

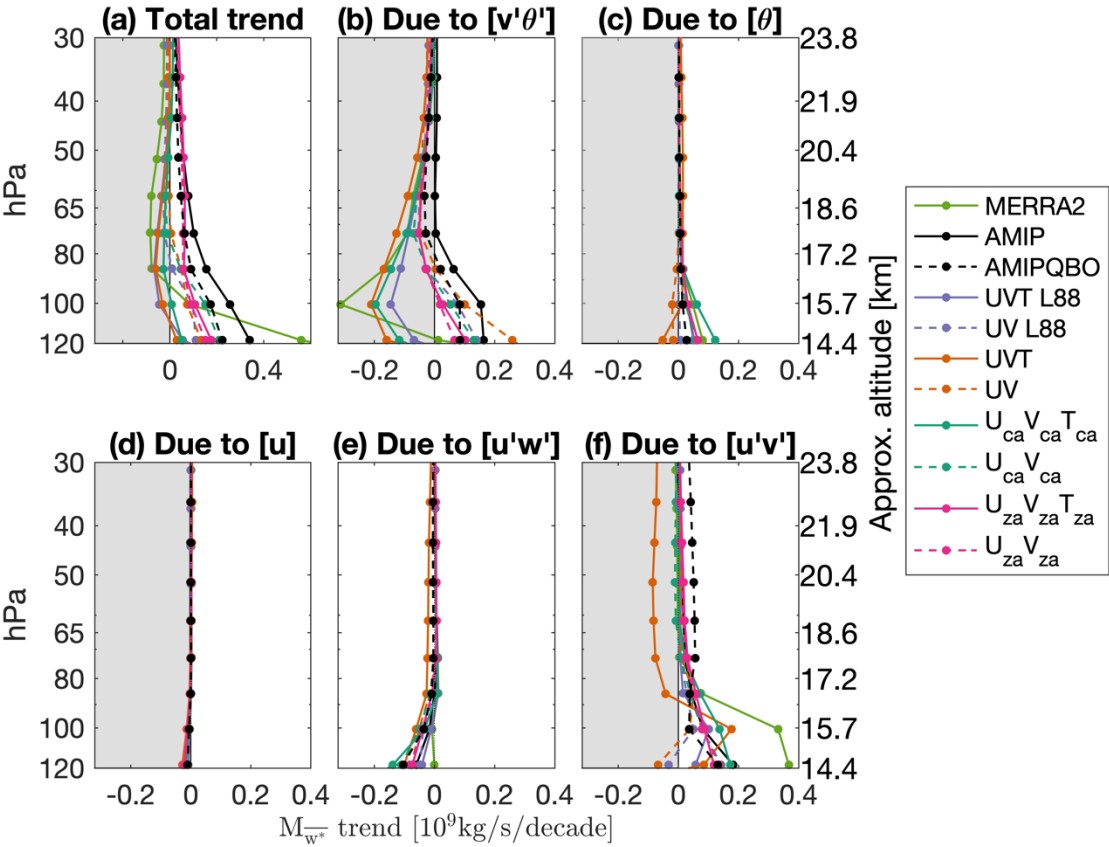

*Figure 8: Tropical upwelling mass flux trend: (a) total trend, (b) trend due to the meridional eddy heat flux, (c) trend due to the zonal-mean potential temperature, (d) trend due to the zonal-mean zonal wind, (e) trend due to the vertical eddy momentum flux, and (f) trend due to the meridional eddy momentum flux. See text for details.*

The climatological Eliassen-Palm flux divergence in the tropics is characterized by an equatorial minimum in negative wave forcing that increases poleward and, in the TTL, downward. In MERRA2 there is a downward consolidation and equatorward shift of the wave forcing at the edge of the pipe in the subtropics from 1980 to 2017, which drives the strong increase in upwelling in the TTL and the minor decrease in upwelling aloft (Fig. 3). Here, the increasing wave drag in the TTL originates from the subtropical lower stratosphere and from within the upwelling region. Contrast this pattern with the pattern observed in the AMIP simulation: enhanced wave propagation from the extratropical troposphere into the subtropical TTL and lower stratosphere, producing acceleration over a deep layer. This pattern resembles the canonical greenhouse gas response observed in CMIP-type simulations [Garcia and Randel 2008, Shepherd and McLandress 2011], and is an enhancement and upward shift of the climatological wave forcing. The AMIPQBO simulation exhibits this pattern only in the Southern Hemisphere.

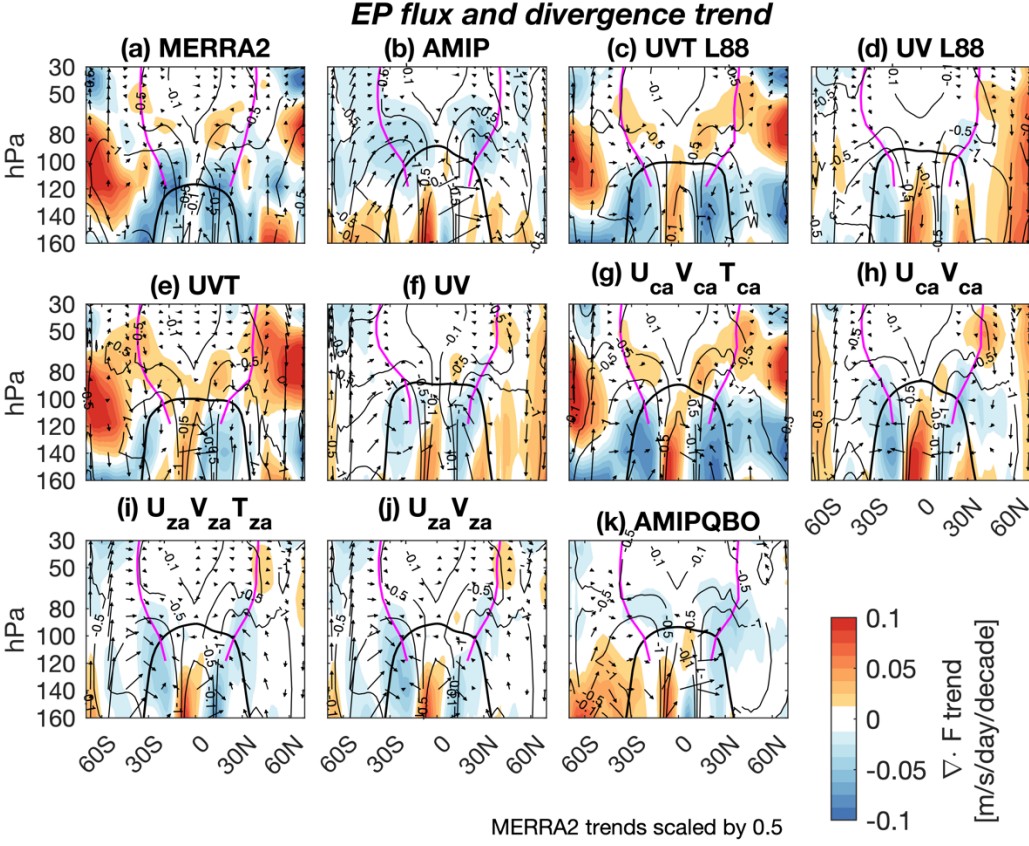

*Figure 9: Eliassen-Palm flux (vectors) and divergence (shading) trends, climatological Eliassen-Palm flux divergence (black contours, -0.1, -0.5, and -1.0 m/s/day), and climatological turn-around latitudes for all simulations (magenta). MERRA2 Eliassen-Palm flux and divergence scaled by 0.5. Eliassen-Palm flux vectors have been scaled to appear consistent with the divergence field, to an arbitrary maximum scale consistent across panels [Edmon et al. 1980]. Tropopause indicated by the thick black line.*

The specified dynamics simulations have patterns that are so similar that they can simply be collapsed into two composites - the UVT L88, UVT, and UVT climatological anomaly simulations, which we will refer to as the "zonal-mean temperature nudging" composite, and the UV L88, UV, UV climatological anomaly, and both zonal anomaly simulations, which we will refer to as the "no zonal-mean temperature nudging" composite (Fig. 10). We emphasize these composites are not merely based on the commonality of this pattern, but also on the binary distinction of whether they nudge zonal-mean temperatures.

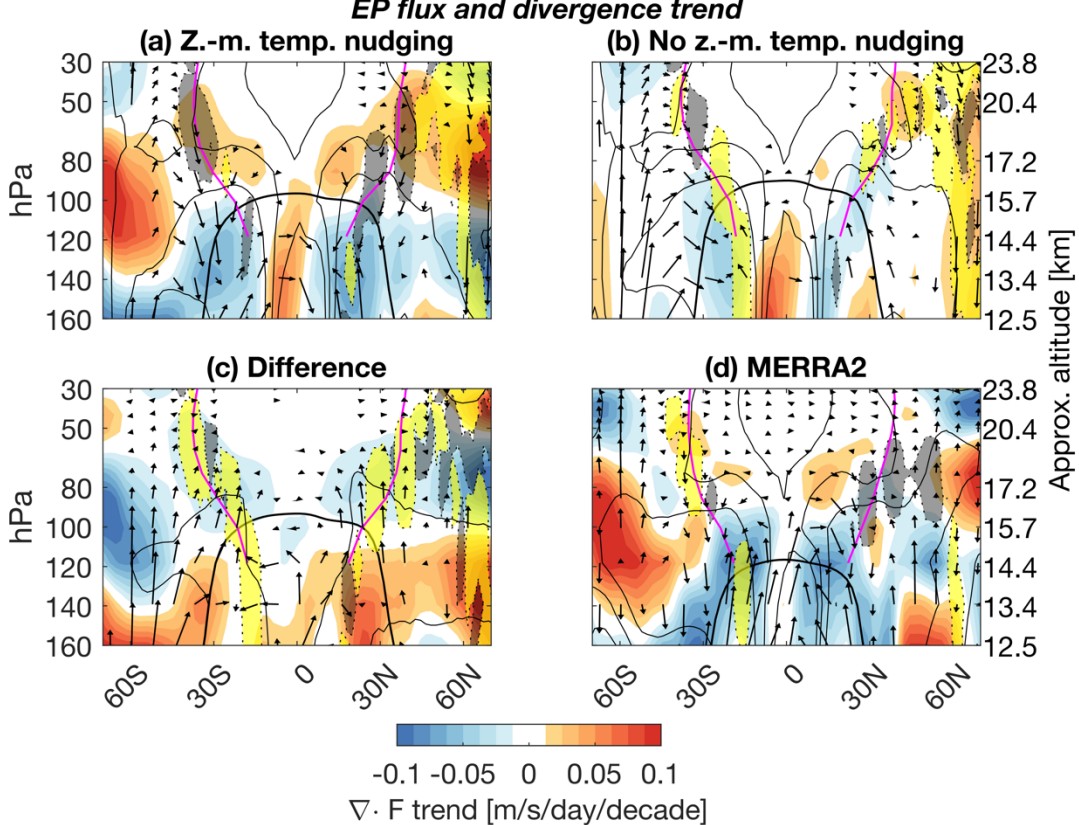

*Figure 10: As in Figure 9, but for the composites of simulations with (a) and without (b) zonal-mean temperature nudging, (c) the difference between the composites, and (d) MERRA2. Also shown are positive (yellow translucence) and negative (black translucence) changes in the index of refraction greater than 1x10⁻¹¹ m⁻².*

In the "zonal-mean temperature nudging" simulations, the change in wave momentum forcing resembles the pattern in MERRA2, but the consolidation of wave forcing from within the pipe to the subtropical lower stratosphere amplifies the climatological drag, rather than shifting it equatorward as in MERRA2. In the Southern Hemisphere in MERRA2, the downward consolidation and equator shift of drag is associated with increases in the index of refraction along the pipe edge between 160 and 100 hPa, while in the "zonal-mean temperature nudging" simulations, there is a decrease in the index in the same location. This suggests a possible mechanism whereby the erroneous decrease in the index of refraction in the "zonal-mean temperature nudging" simulations may be refracting the waves away from the pipe and preventing them from accelerating the upwelling in the TTL. Contrast this with the "no zonal-mean temperature nudging" simulations, in which the positive wave momentum forcing trend in the TTL is due to enhanced propagation from the extratropical troposphere into the lower stratosphere and TTL, similar to the AMIP simulations. However, the waves arc over the subtropics and refract toward the increasing index of refraction, unlike the AMIP simulations and similar to the downward consolidation of wave momentum forcing seen in MERRA2 and the "zonal-mean temperature nudging" simulations. The differenced flux and

divergence trends between the "no zonal-mean temperature nudging" and "zonal-mean temperature nudging" simulations resemble the changes seen in the AMIP simulations. Therefore, the response of the "no zonal-mean temperature nudging" simulations can be understood as the superposition of the "zonal-mean temperature nudging" simulation response - a slightly-incorrect MERRA2 response - and the AMIP response. There is more agreement in the wave momentum forcing changes among the composites and MERRA2 in the Northern Hemisphere, at least at the turn around latitude where it has consequences for tropical upwelling.

## 5 Conclusions and Discussion

Our primary conclusions can be summarized as follows:

1. In WACCM, the particular specified dynamics methodology can have a substantial impact on the climate, trends, and variability of stratospheric upwelling

    a. While specified dynamics schemes that (incidentally) nudge zonal-mean temperatures reproduce variability in tropical stratospheric upwelling, they do not reproduce stratospheric upwelling trends

    b. Specified dynamics schemes that do not nudge zonal-mean temperatures tend to better (but not entirely) reproduce stratospheric upwelling trends, at the expense of variability

2. Nudging the zonally-anomalous circulation tends to most consistently preserve WACCM's climate and reproduce stratospheric upwelling trends in the TTL

3. Gravity wave parameterizations can interfere with a specified dynamics scheme's ability to reproduce upwelling trends

We emphasize we have only assessed these conclusions using WACCM, and have not explicitly examined the impact of the nudging timescale, model resolution, or parameterizations. However, there is no obvious reason why this mechanism should be WACCM-specific. We offer the hypothesis, confirmed here in WACCM, that if there are differences in the climatologies of any nudged model and its input meteorology, upwelling trends will be more poorly reproduced when nudging zonal-mean temperatures than when not nudging to zonal mean temperatures, with the magnitude of error scaling with the difference in the zonal-mean climate.

From these conclusions, we can infer that prescribing zonal-mean MERRA2 temperatures, and in particular the climatological anomalies in zonal-mean temperatures, restricts the temperature response that otherwise spontaneously occurs in the AMIP simulations. However, prescribing the zonal-mean temperatures does not lead to a better reproduction of tropical upwelling trends or wave momentum forcing trends along the pipe edges present in the input meteorology. While we cannot begin to fully answer why this is the case, we hypothesize that it may be due to a mismatch in the climatologies of WACCM and MERRA2.

The "no zonal-mean temperature nudging" simulations, the AMIP simulations, and MERRA2 exhibit similar temperature trends, but the trends in the "zonal-mean temperature nudging" simulations are wildly different, with warming everywhere in the lower stratosphere (Fig. 11). Consider the difference in the climatological temperature between MERRA2 and the AMIP simulation. AMIP is 3 Kelvin colder in the tropical TTL and lower stratosphere, likely due to stronger

adiabatic cooling from the greater upwelling frequency and stronger upwelling mass flux (Fig.'s 1, 2). If the zonal mean temperature is nudged, this will induce a warming tendency in the TTL and lower stratosphere (and a cooling tendency in the upper troposphere; see Weaver et al. [1993] and also Miyazaki et al. [2005] - temperature nudging essentially induces spurious heating). Such a warming tendency would induce a response akin to the westerly-shear QBO phase, with anomalous subsidence on the equator and rising motion in the subtropics (see for example Plumb and Bell [1982], Fig. 1). Indeed, the models in which the zonal-mean temperature is nudged tend to have a reduced upwelling frequency in the equatorial lower stratosphere and TTL (Fig. 2), though the distinctions are less obvious when one considers the upwelling mass flux (Fig. 1).

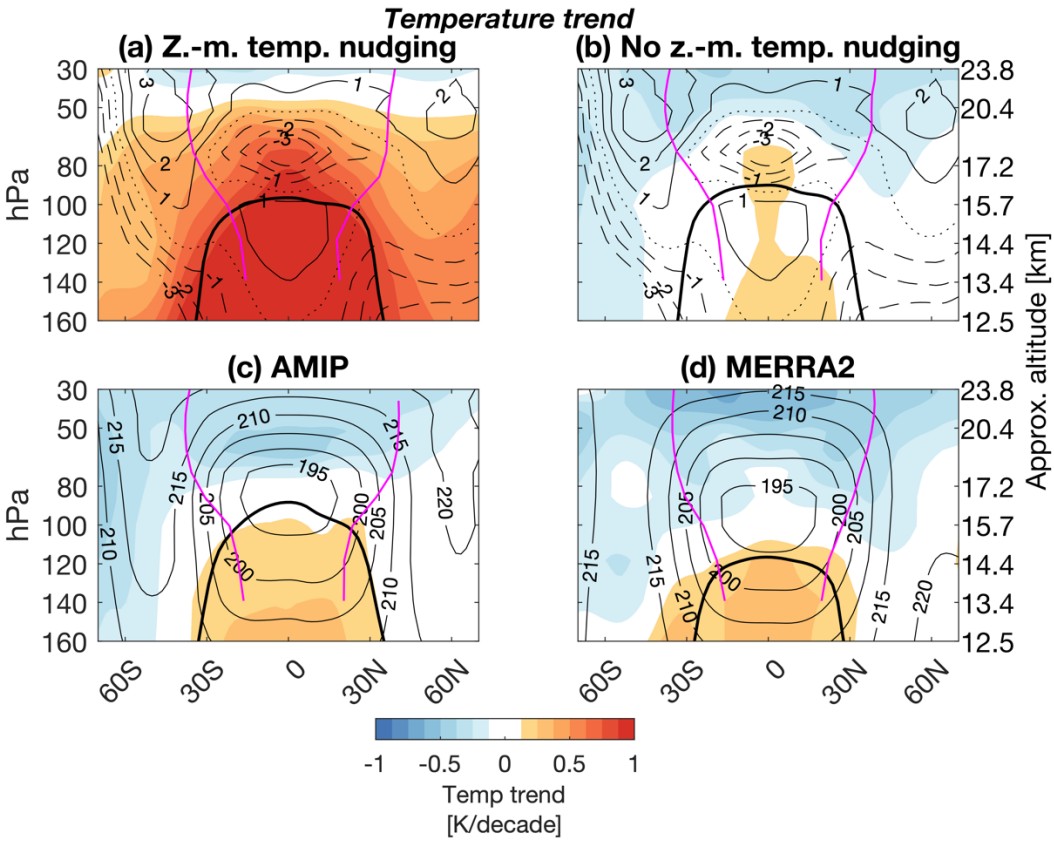

Figure 11: Zonal-mean temperature trends (shading) in the simulations with (a) and without (b) zonal-mean temperature nudging, (c) the AMIP simulation, and (d) MERRA2. Climatological temperatures shown for the AMIP simulation and MERRA2 (black contours, every 5 K), while the climatological temperature difference between the AMIP simulation and MERRA2 is shown on the upper panels (black contours, every 1 K). Also shown are the climatological turn-around latitudes (magenta).

These meridional circulation responses to the nudging tendencies could, in principle, influence trends in potential vorticity which impact wave propagation and ultimately the tropical upwelling mass flux trend (Fig. 10), but a precise mechanism is not obvious. Another possible complication is that differences in the native horizontal resolution between the nudged and meteorological models could lead to non-negligible differences in the wave physics [Boville 1991, Held and Phillipps 1993, Bèguin et al. 2013, Davis and Birner 2016]. While the schemes nudge the eddy terms, the model still has the freedom to determine the final wave generation and breaking processes that lead to zonal-mean wave momentum forcings. Both of these possibilities present potentially serious challenges for specified dynamics schemes. By construction the schemes produce zonal-mean heat and momentum forcings that will engage feedbacks within the model that act to preserve its climatology. Ostensibly, it is not possible in WACCM to nudge zonal-mean temperatures without inducing compensating changes in the model that cause even its zonal-mean temperature trends to diverge rapidly from the trends in the input data (Fig. 11). We should not be surprised, then, that such simulations fail to reproduce tropical upwelling trends, which are driven by nuanced changes in wave propagation and dissipation (Fig.'s 9 and 10).

Parameterized gravity waves also lead to errors between the nudged and meteorological reference simulations [Smith et al. 2017]. Here, they drive deceleration in the TTL and acceleration in the lower stratosphere in WACCM, in part explaining why WACCM tends to have a more positive (negative) upwelling trend in the lower stratosphere (TTL) compared to MERRA2. As long as the parameterization is active, it presents an upper limit on performance in simulating the MERRA2 trends.

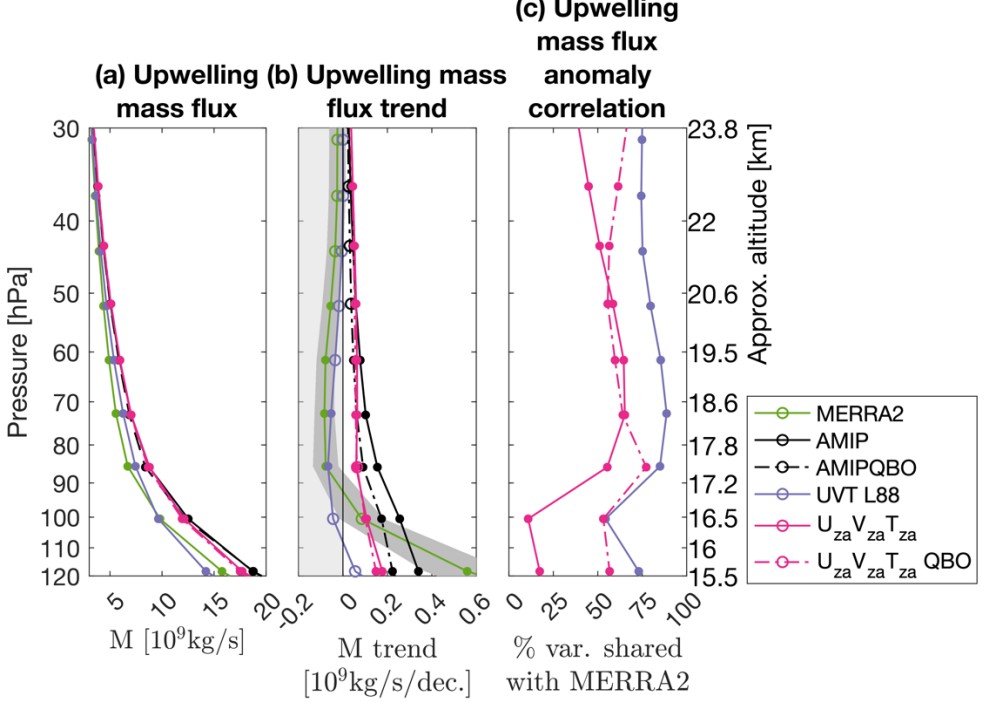

*Figure 12: Climatological (a) upwelling mass flux, (b) upwelling mass flux trend, and (c) anomaly correlation with MERRA2. Shading and symbols in (a) and (b) as in Figures 1 and 3, respectively. $U_{za}V_{za}T_{za}$ QBO is a $U_{za}V_{za}T_{za}$ simulation with zonal-mean zonal wind (QBO) nudging in the tropical stratosphere.*

We have shown that by *not* nudging the zonal-mean temperatures or, even better, *not* nudging zonal-mean variables at all, the simulations are able to respectably reproduce tropical upwelling trends without damaging WACCM's climatology. As the conversion of resolved waves to zonal momentum forcings depends on the zonal mean circulation, the zonal anomaly nudging scheme may be further improved by the inclusion of a separate QBO nudging scheme. Here the combination of QBO nudging (of the zonal-mean zonal winds in the tropics) with zonal anomaly nudging does not impact the climatology or the trends (Fig. 12), but drastically improves the anomaly correlation between WACCM and MERRA2 in the TTL and middle stratosphere to the degree that there is little gain in skill in these regions using the standard UVT L88 scheme. This additional QBO nudging ameliorates the primary drawback of the pure zonal anomaly nudging scheme – its lack of QBO-driven circulation variability. Nevertheless, over this sufficiently-long period, the upwelling trend in the UVT zonal anomaly QBO simulation is almost identical to that in the AMIPQBO simulation, suggesting no gain in skill by constraining historical variability.

In certain situations, nudging to zonal-mean temperatures might be necessary - for example, when there is interest in temperature-dependent chemistry [Froidevaux et al. 2019]. In these cases, the disadvantages of temperature nudging should be investigated and weighed against its advantages. Regardless of the application, model, or input meteorology, care should be taken in interpreting specified dynamics simulations, especially in regard to their modeled trends. Trends are often the small residuals of the balance of large terms; they may be overshadowed by the nudging acting on the climatological differences between the models.

**Data availability**

MERRA2 is provided by NASA's Global Modeling and Assimilation Office at https://gmao.gsfc.nasa.gov/reanalysis/. Source code modifications are archived at doi:10.5281/zenodo.3376232 and can also be provided on request by nadavis@ucar.edu – please note they are compatible with CESM 1.2.2, but may not be compatible with all versions of CESM. Raw and post-processed model output is also available upon request.

**Author contributions**

N. A. Davis, S. M. Davis, R. W. Portmann, and P. Yu designed the model experiments, and N. A. Davis and P. Yu performed the experiments. N. A. Davis produced source code modifications, analyzed model output, and wrote the manuscript with editing and support from all authors.

## Acknowledgements

The National Center for Atmospheric Research is sponsored by the National Science Foundation.. Three anonymous reviewers are thanked for their efforts to provide comments and improve this manuscript. We thank the Global Modeling and Assimilation Office (GMAO) at the NASA Goddard Space Flight Center for producing and distributing the MERRA2 reanalysis, and thank Peter Hitchcock for constructive comments and suggestions.

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
