# Peer review of "A Comprehensive Assessment of Tropical Stratospheric Upwelling in Specified Dynamics CESM1 (WACCM)"

_Geoscientific Model Development, 2019_

## Referee Comment (RC1) · Anonymous Referee #1 · 28 Oct 2019

Overall Comments: In this study the authors explore a broad range of specified-dynamics (SD) simulations in which WACCM is nudged to MERRA-2 meteorological fields in an attempt to quantify and understand the extent to which such SD simulations can reproduce upwelling trends in the underlying reanalysis. Given its implications for, among others, recent investigations into lower stratospheric ozone trends this study is very relevant. It is also an admirable attempt to understand in detail the mechanics of nudging, delving nicely into the momentum balance of upwelling and discrepancies that may arise in these balances among different nudged runs. In this sense the study does stand out as an attempt to address with more rigor than is standard the ways in which nudging can produce non-intuitive trends/variability/etc. For these reasons I

recommend that this paper be accepted with minor revisions. However, there are still several key points that must be addressed. As these are more directed at the delivery and presentation of the results, and not related to fundamental problems that I have with the paper, I have not recommended "major revisions." Nonetheless, they need to be addressed.

My major comments are as follows:

Main Point 1: Throughout the authors argue that it is most desirable to preserve WACCM's free-running climatology (e.g. see discussion at the top of page 8, and various other places). Since this is a not a standard goal of nudging this needs to be better justified. In particular, I find the justification in lines 88-92 unsatisfying. Why is it bad that WACCM-SD reproduce the tropopause (or, more generally, temperature) structure of MERRA-2? Even if that impacts transport isn't that the point? What I would understand is if the authors argued that doing so creates dynamical inconsistencies in the circulation (assuming either the nudging tendency is large enough that it implies spurious vertical velocity analogous to the situation presented in Weaver et al. (1997)). Is this what the authors mean? Given that the use of nudging to climatological means is a central component of this work (and not conventional) I think this needs to be much better explained.

Weaver, Clark J., Anne R. Douglass, and Richard B. Rood. "Thermodynamic balance of three-dimensional stratospheric winds derived from a data assimilation procedure." Journal of the atmospheric sciences 50, no. 17 (1993): 2987-2993.

Main Point 2: I think the authors need to be much more cautious in generalizing the result that zonal mean temperature nudging should not be applied. As the conclusions read (especially point 1 spanning lines 400-404) the authors seem to suggest that this is a general result. However, given that the zonal mean temperature nudging trends are failing to reproduce MERRA-2 through their effects on eddies (via discrepancies in the meridional heat fluxes) I'm highly suspect that other nudging frameworks using

different models (with different balances of resolved vs. parameterized momentum forcing of upwelling) will automatically corroborate these findings. In short, I think the authors need to state clearly how this conclusion depends very specifically on the particular way in which the momentum forcing in WACCM is driving w* and how that may depend on horizontal/vertical resolution and other factors. Of course I notice that line 408-409 seems to direct these questions to future work but this is a bit unsatisfying. If the authors do not wish to do any test simulations (at higher horizontal resolution, for example) they should at minimum be very clear that these results are not likely to be generalizable to other nudging frameworks.

Minor Comments:

Abstract, Ln. 22: I'm a bit confused why the goal is to "preserve WACCM's (free-running) climatology". The whole point of nudging is to draw the free-running model towards the reanalysis in as dynamically consistent a way as possible so I'm not sure why one would want to preserve a (biased) free-running climatological state. I'm sure there's a clear motivation for this but I couldn't identify one in the text (neither here nor in the sections later). See Major Comment 1.

Abstract Ln. 22: "climatological winds" -> Is that also just zonal or meridional too? Ln. 40: What does "the quality of the meteorological data" mean? Please specify.

Ln. 71: Does nudging occur everywhere?

Line 88: This wouldn't happen, though, if one were to nudge "hard" to T (using, for example, a relaxation timescale of a few days, not 50 days). I'm not sure I really understand the point here. Sure, it would change WACCM's tropopause (and other fields) but why is that necessarily a bad thing? Clearly, this would not be good if it were done in such a way that violated dynamical balance but that is more likely to be an artifact of the nudging machinery. What is fundamentally wrong about nudging to the full time-varying reanalysis field?

Line 99: You write that three-hourly MERRA-2 input is used in Line 70 but six-hourly here. Which is it? If six-hourly why was the decision made to coarsen the resolution temporally?

Line 104: Again, can you please justify what you mean by "climatological anomaly nudging scheme is in theory..."? If the nudging was perfect (i.e. converged to assimilation) then it's not obvious to me that there's any fundamental problem with nudging to the full time-varying field.

Line 121: I am assuming other more standard tests have been done (i.e. vertical profile of nudging? changes in nudging timescale?). If so, it should be clarified that these have been done and they have not produced any satisfying simulation in which w* reproduces w* in the underlying reanalysis (here MERRA-2).

Line 168: How did you calculate this from MERRA-2 (as shown in future figures?)? Where did you get all of the components (specifically the subgrid-scale wave momentum forcing)? And which product did you use? You indicated the third hourly fields initially but were six-hourly used here?

Lines 184-193: Again, I am confused. Don't you want to reproduce MERRA-2? See earlier comments. Figure 2 caption: The hatching definition is strange. Per the colorbar definition white contours in all panels should indicate regions where there is upwelling 100% of the time (i.e. fraction of 1). Why doesn't all hatching align with white?

Line 198: Is this frequency calculated daily/monthly/etc. Does the temporal sampling used to evaluate this measure matter?

Line 200: Are you taking w* directly from MERRA-2 or calculating offline in a consistent fashion as for the WACCM simulations? This relates to my earlier question about MERRA-2 mass flux estimates. How exactly are all measures derived from the MERRA-2 output?

Line 201: What if you just compare climatological annual mean w* between WACCM

and MERRA-2? That's more standard – does that show the same sort of difference (i.e. w* smaller in MERRA-2)? I find this "split" in upwelling frequency in MERRA-2 curious only because it doesn't appear to manifest in the climatology of w* (see Figure 10-3 in Bosilovich et al. (2015)). Note that in MERRA this region of anomalous downwelling was present but it was corrected in MERRA-2. This seems to be at odds with what the current study is showing. Can the authors explain this discrepancy? The easiest thing to do would be just to plot the climatology and see if you can reproduce the aforementioned figure...

Bosilovich, M. G. (2015). MERRA-2: Initial evaluation of the climate. National Aeronautics and Space Administration, Goddard Space Flight Center. Available at https://gmao.gsfc.nasa.gov/pubs/docs/Bosilovich803.pdf

Lines 248:251: So this is a really important conclusion – the lack of any convergence of the trends to MERRA-2 in Figure 4 is striking (and frustrating!). This is a merely a comment that I like this figure. Line 280: Indeed. Hence, why is this the primary goal of the paper? Again, more justification needed. See earlier comments.

Line 313: Given the larger role played by the (parameterized) GWD in contributing to upwelling trends in WACCM does this imply that your conclusions will depend largely on horizontal and vertical resolution? One would think that as more of the waves contributing to w* are resolved then the disparities with MERRA-2 (in terms of the physical mechanisms forcing the trends) will get smaller. Have you looked at SD simulations at different horizontal resolutions?

Line 357: You can see that enhanced wave propagation clearly in the AMIP run but not so clearly in the AMIPQBO run (no evidence in NH extratropics)...please check.

---

## Referee Comment (RC2) · Anonymous Referee #2 · 11 Nov 2019

**Anonymous Reviewer 2 (Comments):**

**Review of "A Comprehensive Assessment of Tropical Stratospheric Upwelling in Specified Dynamics CESM1.2.2 (WACCM)" by Nicholas A. Davis et al. (2019)**

**General Comments**

The authors of this study probe the impacts of nudging WACCM towards MERRA-2 meteorology in a number of Specified-Dynamics (SD) simulations. The novelty of this study lies in exploring discrete nudging "flavours" in order to determine the degree in which they can accurately simulate the mean state, the interannual variability and the upwelling trends of the residual circulation seen in MERRA-2 reanalysis product. This is a commendable attempt to understand how the implementation of various nudging frameworks affects the self-consistency of a chemistry-climate simulation highlighting the possible implications regarding the accuracy of transport processes associated mainly with ozone trends in the lower stratosphere as seen in recent studies. Apart from the question of which nudging scheme appears to better reproduce MERRA-2 (and to a second degree the free-running WACCM) spatial and temporal upwelling characteristics, the study comprehensively investigates the attribution of these discrepancies by shedding some light on the physical drivers of the upwelling trends. Nudging acts as an additional non-physical tendency in the model equations and it is quite important to evaluate their potential artificial effects on the model dynamics. Therefore, this study greatly improves the understanding regarding the degree of impact arising from the choice (or not) of a particular nudging scheme, albeit in a single model framework. For the reasons above, I recommend this study to be accepted and published with minor revisions. There are a few points that I think they should be addressed by the authors, more to do with enhancing the introduction of the paper by adding a substantial amount of discussion on the nudging studies literature.

**Specific comments**

**1.** I find the introduction to be relatively short and lacking in terms of literature related to nudging studies. In order to set the scene better and highlight that nudging studies are not just used for lower stratospheric ozone trends, more references would be extremely valuable to the reader. One of the first attempts to obtain a comparison between a GCM relaxed towards analyses and the analyses themselves is detailed in Jeuken et al. (1996). There are multiple studies looking at specific meteorological events, such as van Aalst et al. (2004) looking into the Arctic winter transport processes at the end of the 20th century or the SSW during 2009-2010 winter in Akiyoshi et al. (2016). Similarly, nudged simulations were used to focus on the effects of volcanic eruptions on stratospheric tracers such as water vapour in Loffler et al. (2016), to

infer the global-mean volcanic effective radiative forcing over the satellite era in Schmidt et al. (2018), as well as to estimate the chemical effects of monsoon circulations on volcanic sulphur particles seen in Solomon et al. (2016). Additionally, Solomon et al. (2015), studied the polar ozone depletion in 2011, using a nudged version of WACCM. In fact, the latter three studies, used a previous version of CESM1-WACCM which is nudged towards an older version of the reanalysis (MERRA), with and without nudging the temperature respectively. A bit of discussion regarding the differences between the nudging schemes used in the aforementioned studies and in the current study certainly wouldn't hurt. I would also recommend some discussion with respect to the differences in between a CTM and an CCM-SD model run as nudging, especially when trying to interpret the differences between Ball et al. (2018) and Chipperfield et al. (2018). CTMs are forced directly with the full 3-D circulation from reanalyses and after many years of optimizations they have been proven quite successful at simulating stratospheric tracers on various timescales (Chipperfield, 1999; Mahieu et al. 2014). On the other hand, CCMs are much more recent tools and exhibit deep-rooted differences when compared to a CTM when looking at their tracer advection, and some discussion regarding their differences would be also a good idea.

**2.** There is a hint in the study that the overarching aim of the study is to capture the free-running WACCM climatology rather than the climatology of MERRA2 - some justification is required if that is the case. Although the research question (reproducing the residual circulation variability and trends of MERRA2) of the study is clearly stated in the introduction (lines 56-57), there are various places across the text where the message appears to be the reproduction of free-running WACCM. As an example, it is stated that the nudging will create conflicts due to the differences between WACCM and MERRA2 underlying climatologies (explained in section 2 - lines 88-92). I would suggest rephrasing the relevant parts where the point seems to be lost. The point of nudging is exactly to reproduce the reanalyses themselves albeit exhibiting spurious features in the stratospheric residual circulation. It should be noted that between reanalysis products and among different estimates of wbar* there lies significant uncertainty with respect to upwelling trends as seen in Abalos et al. (2015) as well as Kobayashi and Iwasaki, (2016).

3. There's no mention of the MERRA2 output the authors have used throughout this study. How did you calculate the TEM diagnostics? Did you perform the calculation on the native MERRA2 levels? What output have you used? How about its temporal frequency? This information needs to be included by describing all the above in either section 2 or section 3.

**Minor comments**

**Line 22** : See discussion above (point 2)

**Line 55** : Some more references are needed here such as Abalos et al. (2015) and Kobayashi and Iwasaki, (2016) with findings related to the discrepancies in the trends of the residual circulation between reanalysis.

**Line 62** : In the context of this study is quite clear that nudging zonal-mean temperatures alters the meridional eddy momentum and heat fluxes in the TTL without being successful in simulating the underlying MERRA2 trends. Applying a thermal nudging (temperature) could potentially lead to a sustained spurious heat source in the model, which leads to a stronger BDC in the lower stratosphere as seen in Miyazaki et al. (2005) with a different model. However, the last sentence is quite strong as a statement and generalizes a result which is model specific. Therefore, I would recommend rephrasing this bit so it doesn't strike as misleading.

**Line 69** : When nudging, the choice of the relaxation timescale can play an important role (Merryfield et al. 2013), although there is no consensus that a specific timescale necessarily leads to an improvement (Hardiman et al. 2017). I'm aware of a standard full WACCM-SD CCMI simulation using a nudging timescale of 5 hours - towards MERRA though (Orbe et al. 2018; Orbe et al. 2019 under review in ACPD), have you performed any additional runs with this timescale (or have plans) to compare with?

**Lines 84-87** : It would be very helpful (at least to me) if you could include a figure (in the supplement) showing the vertical profiles of the pressure levels in WACCM-L66/L88 and MERRA2 to better highlight their spacing differences throughout the depth of the atmosphere.

**Lines 98-99** : Here you mention that 6-hourly MERRA2 anomalies are used for nudging WACCM, I assume you are interpolating in time to nudge every 3 hours (see line 71)?

**Line 105** : Please clarify that this improvement refers specifically to aerosol-climate interactions in Zhang et al. (2014).

**Lines 155 & 159** : Please correct the reference - it is : Hardiman et al. (2010)

**Line 164** : SF6 can be considered linear only by approximation, characterised by a fast growth rate and there needs to be a correction for this. See Garcia et al. (2011).

**Line 224** : This line refers to Figure 3b, where you calculate the difference in the trends compared to MERRA2. For clarity, it would be better to rephrase and not use the word negative but either smaller or bigger to reflect their differences. E.g. in the TTL all WACCM runs + MERRA 2 have positive trends as seen in Figure 3a, and the term negative might be misleading as the trends are just smaller (but still positive).

**Line 228** : " *The standard UVT...*" - Clarify that this holds true for both versions (L66 + L88).

**Lines 241 - 252** : Excellent discussion (and figure)!

**Line 309** : By gravity wave (GW) momentum forcing are you referring to all the parameterizations? Meaning orographic (OGW) + non-orographic (NOGW) gravity wave drag put together? Please clarify.

**Line 311** : McLandress and Shepherd (2009), using CMAM, show the total contribution of both resolved and parameterized wave drag occuring at the edge of the pipe in the lower stratosphere in boreal winter in their figure 18. However, this lumps together resolved (major contributor at the edge of the pipe) and all GW parameterizations while the orographic gravity wave drag contributes more in the NH mid-latitudes instead. I would suggest caution drawing parallels to this result which remains model specific. Different parameterizations lead to various magnitudes of contributions to the upwelling throughout the stratosphere and specifically for the versions of CMAM over the past decade, it has been shown that the NOGW contributes negatively in the upwelling in the lower stratosphere in SPARC, 2010 and more recently in Chrysanthou et al. (2019).

---

## Referee Comment (RC3) · Anonymous Referee #3 · 19 Nov 2019

Many studies have been performed whereby otherwise freely-running chemistry-climate models have had the day-to-day evolution of the dynamical fields constrained to follow the historical evolution as represented by reanalysis datasets. Here, Davis et al. present an analysis of different nudging schemes, using different combinations of variables or only nudging to zonal anomalies that are calculated in different ways, to assess the impact on the residual circulation of the lower stratosphere with a particular emphasis on how the nudged simulations differ with freely-running simulations and with the reanalysis dataset used for nudging. The study is very nicely performed and includes a convincing mechanistic diagnosis of the ways in which nudging of different variables affects the trends in tropical upwelling.

[Figure]

I really have no major concerns on the methodology or analysis presented here and my comments are mostly minor. One concern I do have, however, is the presentation of the effects of nudging zonal mean temperature on reproducing trends. In the abstract, at lines 21 – 23, the authors state that nudging to anomalies better reproduces trends in stratospheric upwelling, period. Taking a broader view, it would seem that nudging anomalies produces trends in upwelling that are more similar to the trends produced by the free-running (AMIP) simulation. This is clearly shown in Figure 3a, where the schemes that involve nudging to anomalies are much closer to the free-running simulation both in the TTL and in the lower stratosphere. As a consequence of the trends produced by the free-running AMIP simulation, the simulations nudged to zonal anomalies agree better with MERRA2 in the TTL but agree more poorly through the lower stratosphere. The degree of differences to the AMIP simulation across different nudging schemes also extends to the analysis of the EP-flux trends where it is stated (lines 392 - 394) that 'the response of the "no zonal-mean temperature nudging" simulations can be understood as the superposition of the "zonal-mean temperature nudging" simulation response - a slightly-incorrect MERRA2 response – and the AMIP response.' From both the analysis of trends and the analysis of the mechanism it appears that the response of schemes that do not affect the zonal-mean temperature produce trends that are more like the AMIP free-running simulation. I would suggest the authors should not overstate the conclusions of the effects of nudging temperature on the ability of the nudged model to reproduce the trends in the reanalysis as it would seem to depend significantly on the underlying behaviour of the free-running simulation.

Not at all a criticism, but more of a puzzled commentary. Figure 11 shows that nudging the zonal mean of temperature from MERRA produces temperature trends that disagree with the trends in MERRA. I can accept that the cause of the differences in the stratophere are not fully understood and may be related to unintended secondary circulations, but the anomalous trends found only in simulations that nudged the zonal mean temperature extend deep into the troposphere. In fact, the trends in the upper

tropical troposphere appear to be three or four times larger than the trends in the same region found in MERRA2. Do you have any explanation for the discrepancy in trends in the troposphere and could there be links to the trends in the lower stratosphere?

Minor comments

Lines 98 – 99, for the case where WACCM is nudged towards anomalies it is stated 'To generate the nudging input, 6-hourly MERRA2 U, V, and T anomalies are calculated...' but a bit earlier, at lines 68 – 71, when the default nudging scheme is described it is stated that the MERRA2 reanalysis is supplied to the model every 3 hours. Is this difference real or just a typo? And if it is real, have the authors considered the differences in model behaviour that may be caused by reducing the frequency by a factor of two? Part of the motivation behind pointing this out is an open question about the effect of linearly interpolating in time between the available reanalysis.

Line 180 – Figure 1, I might suggest reducing the vertical extend to maybe 5 hPa so that the horizontal scale can be expanded. None of the other graphs extend beyond 30 hPa.

Line 216 – missing 'A' in 'MERR2'

Lines 239-240: Here it is stated that 'This all demonstrates that (incidentally) nudging zonal-mean MERRA2 temperatures has a negative impact on the upwelling trend morphology and magnitude.' I see how the findings of the correlation coefficient of trends with MERRA2 being largest for simulations that do not nudge the zonal-mean temperature supports the statement on morphology. But the magnitude of the trend over large regions of the vertical profile shown in Figure 3 is closest to MERRA2 for the simulations that do nudge zonal temperature. The magnitude of the trends in UVT is closer to MERRA2 than UV, and UVT(ca) is closer than UV(ca) between 90 hPa and 40 hPa. While the magnitude of the trends in UVT(za) are the furthest from MERRA2 everywhere above 90 hPa. The experiments where zonal average temperature is not nudged are closer to the AMIP simulation and this is an advantage in the TTL as the

AMIP simulation has the largest positive trends and is thus closest to MERRA2. But producing trends closer to the freely-running AMIP simulation becomes a disadvantage higher up where the freely-running AMIP simulation produces more positive trends than MERRA2. Having read a bit further, I see how you eventually address this (and I particularly like Figure 4) but the statement at Lines 239-240 about the effect of nudging zonal-mean temperature on the magnitude of trends seems unsupported.

Lines 284 - 285: It is stated here that the poor performance of the zonal anomaly nudging in reproducing variability in upwelling below 85 hPa suggests 'a strong role for the zonal-mean circulation in transforming wave dynamics into zonal-mean momentum forcing and therefore upwelling (Fig. 6).' Are you suggesting that the QBO has a role to play in upwelling in the TTL? Is there anything to be seen correlating the MERRA2 variability with that of the AMIPQBO run? [ Okay, way down at Line 460 I see where you address the role of the QBO on variability in the TTL using UVT(za) nudging. ]

Line 363 – 366 – The caption for Figure 9 does not mention what is indicated by the thick black line. Is it the lapse rate tropopause?

Line 414 – minor typo on '\hypothesize'

―――――――――――――――――

---

## Author Comment (AC1) · 21 Dec 2019

**Response to Referee #1**

*Overall Comments: In this study the authors explore a broad range of specified-dynamics (SD) simulations in which WACCM is nudged to MERRA-2 meteorological fields in an attempt to quantify and understand the extent to which such SD simulations can reproduce upwelling trends in the underlying reanalysis. Given its implications for, among others, recent investigations into lower stratospheric ozone trends this study is very relevant. It is also an admirable attempt to understand in detail the mechanics of nudging, delving nicely into the momentum balance of upwelling and discrepancies that may arise in these balances among different nudged runs. In this sense the study does stand out as an attempt to address with more rigor than is standard the ways in which nudging can produce non-intuitive trends/variability/etc. For these reasons I recommend that this paper be accepted with minor revisions. However, there are still several key points that must be addressed. As these are more directed at the delivery and presentation of the results, and not related to fundamental problems that I have with the paper, I have not recommended "major revisions." Nonetheless, they need to be addressed.*

*Main Point 1: Throughout the authors argue that it is most desirable to preserve WACCM's free-running climatology (e.g. see discussion at the top of page 8, and various other places). Since this is a not a standard goal of nudging this needs to be better justified. In particular, I find the justification in lines 88-92 unsatisfying. Why is it bad that WACCM-SD reproduce the tropopause (or, more generally, temperature) structure of MERRA-2? Even if that impacts transport isn't that the point? What I would understand is if the authors argued that doing so creates dynamical inconsistencies in the circulation (assuming either the nudging tendency is large enough that it implies spurious vertical velocity analogous to the situation presented in Weaver et al. (1997)).Is this what the authors mean? Given that the use of nudging to climatological means is a central component of this work (and not conventional) I think this needs to be much better explained.*

*Weaver, Clark J., Anne R. Douglass, and Richard B. Rood. "Thermodynamic balance of three-dimensional stratospheric winds derived from a data assimilation procedure."Journal of the atmospheric sciences 50, no. 17 (1993): 2987-2993.*

We apologize for being unclear and stating our goal before its motivation. We suspected that a key factor driving the inability of WACCM to produce the proper upwelling trends were disagreements in the climatologies of the meteorological input

and WACCM. One can imagine it's easy for a model to have its variability nudged - but to nudge the mean could incur substantial undesired responses in the model, exactly analogous to the idea presented in Weaver et al. [1993] and some other references suggested by other reviewers.

We now explicitly state in the introduction on lines 78-81 that "Given that multidecadal trends in the earth system tend to be the residual of a balance of much larger terms, we hypothesize that disagreements between the climatologies of the input meteorology and the nudged model may lead to spurious circulations that interfere with upwelling trends."

We have also cleaned up the manuscript by revising any mention of "preserving WACCM's climatology" to more accurately reflect that this is not an end goal in itself, but rather a hypothesis that doing so may improve the upwelling trends in the TTL and lower stratosphere.

Weaver et al. [1993] is now cited in our discussion of spurious heating and temperature trends on line 519.

*Main Point 2: I think the authors need to be much more cautious in generalizing the result that zonal mean temperature nudging should not be applied. As the conclusions read (especially point 1 spanning lines 400-404) the authors seem to suggest that this is a general result. However, given that the zonal mean temperature nudging trends are failing to reproduce MERRA-2 through their effects on eddies (via discrepancies in the meridional heat fluxes) I'm highly suspect that other nudging frameworks using different models (with different balances of resolved vs. parameterized momentum forcing of upwelling) will automatically corroborate these findings. In short, I think the authors need to state clearly how this conclusion depends very specifically on the particular way in which the momentum forcing in WACCM is driving w\* and how that may depend on horizontal/vertical resolution and other factors. Of course I notice that line 408-409 seems to direct these questions to future work but this is a bit unsatisfying.If the authors do not wish to do any test simulations (at higher horizontal resolution, for example) they should at minimum be very clear that these results are not likely to be generalizable to other nudging frameworks.*

We agree that the summary of results gave the impression we were prescribing a set of best practices generally, and not just for WACCM. We have added "In WACCM" to conclusion #1 on line 486.

However, we disagree that this result isn't likely to be generalizable to any other nudging simulation at a different resolution or in a different model. There is certainly some dependence of the resolved wave field on resolution, but it does not really change the dynamical regime or dominant balance of terms [Boville 1991; Held and Phillipps 1993; Béguin et al. 2013; Davis and Birner 2016]. The likelihood that different resolutions/models produce substantially different dynamical balances when averaged over almost half of the earth seems unlikely.

We do not have a robust mechanism that explains why the full nudging cannot produce the correct upwelling trends in the TTL/lower stratosphere. However, we have shown that when we craft the nudging scheme so that it does not nudge the climatological mean, or the climatological nor zonal mean, and temperature in particular, the upwelling trends are in better agreement with the input meteorology.

We have expanded our discussion on lines 496-501 to note our hypothesis moving forward. It is left as future work, for either us or other authors, to assess whether it is true because it is well outside the scope of this paper.

"We emphasize we have only assessed these conclusions using WACCM, and have not explicitly examined the impact of the nudging timescale, model resolution, or parameterizations. However, there is no obvious reason why this mechanism should be WACCM-specific. We offer the hypothesis, confirmed here in WACCM, that if there are differences in the climatologies of any nudged model and its input meteorology, upwelling trends will be more poorly reproduced when nudging zonal-mean temperatures than when not nudging to zonal mean temperatures, with the magnitude of error scaling with the difference in the zonal-mean climate."

**Minor Comments:**

*Abstract, Ln. 22: I'm a bit confused why the goal is to "preserve WACCM's (free-running) climatology". The whole point of nudging is to draw the free-running model towards the reanalysis in as dynamically consistent a way as possible so I'm not sure why one would want to preserve a (biased) free-running climatological state. I'm sure there's a clear motivation for this but I couldn't identify one in the text (neither here nor in the sections later). See Major Comment 1.*

See our response to your main comment.

*Abstract Ln. 22: "climatological winds" -> Is that also just zonal or meridional too?*

We are not sure what you mean by "also just zonal", by not specifying "zonal" or "meridional" we thought "climatological winds" would imply the full horizontal wind field. We are happy to address this if you can clarify.

*Ln.40: What does "the quality of the meteorological data" mean? Please specify.*

This is a suggestion from Ball et al. [2018] as a possible contributor to errors in nudging schemes. Reanalyses are known to have difficulty conserving mass, momentum, and heat, so nudging can introduce inconsistencies that the model must find a way to balance. See our further discussion on lines 74-76.

*Ln. 71: Does nudging occur everywhere?*

We note on lines 107-109: "In this configuration, the model instead runs on 88 levels - 72 levels from the surface to the lower mesosphere, on MERRA2 hybrid levels, with a further 16 free-running levels in the upper atmosphere." We have also added two plots to the supplemental information detailing the MERRA2 levels. The log-scale plot illustrates the lid of MERRA2 and the 16 free-running levels above.

*Line 88: This wouldn't happen, though, if one were to nudge "hard" to T (using, for example, a relaxation timescale of a few days, not 50 days). I'm not sure I really understand the point here. Sure, it would change WACCM's tropopause (and other fields) but why is that necessarily a bad thing? Clearly, this would not be good if it were done in such a way that violated dynamical balance but that is more likely to be an artifact of the nudging machinery. What is fundamentally wrong about nudging to the full time-varying reanalysis field?*

Any nudging term is an unphysical quantity; by construction it violates any momentum, heat, or mass balance in the model. Given that, hard nudging introduces even larger unphysical terms than weak nudging.

Nudging variability about the mean is probably not too unphysical, given that it is a temporary departure from the mean, but nudging the mean itself will engage the processes that set the modeled climate in the first place.

One way to limit the unphysical tendencies is to lengthen the nudging timescale, but eventually the model will no longer actually reproduce variability. We instead took a different approach to try to see which physical field was most responsible for the inability of WACCM to reproduce upwelling trends in the TTL and lower stratosphere.

*Line 99: You write that three-hourly MERRA-2 input is used in Line 70 but six-hourly here. Which is it? If six-hourly why was the decision made to coarsen the resolution temporally?*

This was a typo, it is 3-hourly in all cases.

*Line 104: Again, can you please justify what you mean by "climatological anomaly nudging scheme is in theory..."? If the nudging was perfect (i.e. converged to assimilation) then it's not obvious to me that there's any fundamental problem with nudging to the full time-varying field.*

See our response to your main comment #1, and also the response to the specific comment on line 88. We have cleaned up the manuscript so it is clear that our hypothesized way to improve the trends is to preserve the climate of WACCM.

*Line 121: I am assuming other more standard tests have been done (i.e. vertical profile of nudging? changes in nudging timescale?). If so, it should be clarified that these have been done and they have not produced any satisfying simulation in which w\* reproduces w\* in the underlying reanalysis (here MERRA-2).*

We think it's fair to argue that a study performing standard tests might not add as much to the literature as one performing more novel tests. Our goal was to determine which physical fields are most impactful, not the strength of the nudging.

We did investigate the timescale issue, but to get the timescale short enough to sample the phase space (for example, 12 hours) we had to decrease the physics timestep. This presented a conundrum, because shortening the physics timestep can change how the convective and gravity wave schemes impact the circulation. We now note on lines 92-96:

"We attempted to run WACCM at up to 10% per timestep (or, 5 hour timescale), but this required increasing the physics parameterization sub-cycling due to convective scheme errors - the "nsplit" parameter. Such simulations are not numerically

comparable so we have chosen to avoid assessing the impact of nudging timescale, though it is known to have varied impacts [Merryfield et al. 2013, Hardiman et al. 2017, Orbe et al. 2017]."

The biggest constraint on the scope of this study was the amount of computing resources available to us. Repeating experiments at different timescales or resolutions was just not feasible.

*Line 168: How did you calculate this from MERRA-2 (as shown in future figures?)? Where did you get all of the components (specifically the subgrid-scale wave momentum forcing)? And which product did you use? You indicated the third hourly fields initially but were six-hourly used here?*

Thanks for this, we should have noted which product we used. We now state on lines 96-97 "WACCM is nudged toward the MERRA2 reanalysis instantaneous assimilation ("ASM") product", and state on lines 198-199, "Eddy fluxes are calculated every 3-hourly output interval in MERRA2 on native levels, while eddy fluxes are output as a monthly-mean value in WACCM".

Regarding the subgrid-scale forcings, we now note on line 199, "We use averaged output for zonal-means and gravity wave tendencies", to clarify that instantaneous fields are only used to calculate the eddy fluxes. As gravity wave tendencies can be highly variable in time, an average over the instantaneous values would not be accurate like a averaged output.

*Lines 184-193: Again, I am confused. Don't you want to reproduce MERRA-2? See earlier comments.*

Not necessarily, no, see our response to your main comment #1.

*Figure 2 caption: The hatching definition is strange. Per the colorbar definition white contours in all panels should indicate regions where there is upwelling 100% of the time (i.e. fraction of 1). Why doesn't all hatching align with white?*

The colorbar indicates any region with upwelling >= 90% of the time will be white; the hatching therefore is used to indicate the exceptional areas where there is exactly 100% upwelling.

*Line 198: Is this frequency calculated daily/monthly/etc. Does the temporal sampling used to evaluate this measure matter?*

As we note in the methods section, all values examined in this study are monthly-means. We are sure that the temporal sampling matters - in the annual mean there will clearly be upwelling everywhere in the tropics, but that washes out the variability we are interested in.

*Line 200: Are you taking w\* directly from MERRA-2 or calculating offline in a consistent fashion as for the WACCM simulations? This relates to my earlier question about MERRA-2 mass flux estimates. How exactly are all measures derived from the MERRA-2 output?*

See our response to your comment on line 168.

*Line 201: What if you just compare climatological annual mean w\* between WACCM and MERRA-2? That's more standard – does that show the same sort of difference (i.e. w\* smaller in MERRA-2)? I find this "split" in upwelling frequency in MERRA-2 curious only because it doesn't appear to manifest in the climatology of w\* (see Figure 10-3 in Bosilovich et al. (2015)). Note that in MERRA this region of anomalous downwelling was present but it was corrected in MERRA-2. This seems to be at odds with what the current study is showing. Can the authors explain this discrepancy? The easiest thing to do would be just to plot the climatology and see if you can reproduce the aforementioned figure…*

*Bosilovich, M. G. (2015). MERRA-2: Initial evaluation of the climate.NationalAeronautics and Space Administration, Goddard Space Flight Center. Available at https://gmao.gsfc.nasa.gov/pubs/docs/Bosilovich803.pdf*

Upwelling frequency (Fig. 2) is a complementary measurement to the upwelling mass flux (Fig. 1), and provides useful information about the permanency of upwelling at any location - something the average vertical velocity cannot describe.

See the plot below. The annual-mean w\* is similar between the two. This suggests that MERRA2 often has periods of slight downwelling on the equator in the lower stratosphere, but small enough that it still has net upwelling in the annual mean.

*Lines 248:251: So this is a really important conclusion – the lack of any convergence of the trends to MERRA-2 in Figure 4 is striking (and frustrating!). This is a merely a comment that I like this figure.*

We agree this is important; another reviewer has suggested we describe the behavior of the nudging schemes as making the trends more or less AMIP-like, which can be easily seen in this figure.

*Line 280: Indeed. Hence, why is this the primary goal of the paper? Again, more justification needed. See earlier comments.*

See our response to your main comment #1; primarily because nudging to the full meteorology produces the wrong sign of the upwelling trend in the TTL, which probably influenced the conclusions of the Ball et al. [2018] paper which claimed models could not explain the reduction in lower stratospheric ozone seen in observations.

*Line 313: Given the larger role played by the (parameterized) GWD in contributing to upwelling trends in WACCM does this imply that your conclusions will depend largely on horizontal and vertical resolution? One would think that as more of the waves contributing to w\* are resolved then the disparities with MERRA-2 (in terms of the physical mechanisms forcing the trends) will get smaller. Have you looked at SD simulations at different horizontal resolutions?*

It is possible that the GWD may drive different trends at higher vertical resolution - for example, at the 110-level WACCM in CESM2 that generates a spontaneous QBO. But the intermodel spread in the upwelling trends is due to resolved wave drag. We would not expect that over the range of reasonable horizontal resolutions, say 2.8 vs. 1 degree, that we will resolve substantially more convective gravity waves, which are dominant in the tropics - their scales are on the order of the latent heating within deep convective clouds, which is orders of magnitude smaller in scale.

*Line 357: You can see that enhanced wave propagation clearly in the AMIP run but not so clearly in the AMIPQBO run (no evidence in NH extratropics)...please check.*

Thanks, this evaded us. On lines 444-445 we now state "The AMIPQBO simulation exhibits this pattern only in the Southern Hemisphere."

**References**

Béguin, A., O. Martius, M. Sprenger, P. Spichtinger, D. Folini, and H. Wernli, 2013: Tropopause level Rossby wave breaking in the Northern Hemisphere: A feature-based validation of the ECHAM5-HAM climate model. *Int. J. Climatol.*, 33, 3073-3082, doi: 10.1002/joc.3631.

Boville, B. A., 1991: Sensitivity of simulated climate to model resolution. *J. Climate*, 4, 469-485, doi:10.1175/1520-0442(1991)004,0469:SOSCTM.2.0.CO;2.

Davis, N. and T. Birner, 2016: Climate model biases in the width of the tropical belt. *J. Climate,* 29, 1935-1954, https://doi.org/10.1175/JCLI-D-15-0336.1.

Held, I. M., and P. J. Phillipps, 1993: Sensitivity of the eddy momentum flux to meridional resolution in atmospheric GCMs. *J. Climate*, 6, 499-507, doi: 10.1175/1520-0442(1993)006,0499:SOTEMF.2.0.CO;2.

---

## Author Comment (AC2) · 21 Dec 2019

**Response to Referee #2**

*The authors of this study probe the impacts of nudging WACCM towards MERRA-2 meteorology in a number of Specified-Dynamics (SD) simulations. The novelty of this study lies in exploring discrete nudging "flavours" in order to determine the degree in which they can accurately simulate the mean state, the interannual variability and the upwelling trends of the residual circulation seen in MERRA-2 reanalysis product. This is a commendable attempt to understand how the implementation of various nudging frameworks affects the self-consistency of a chemistry-climate simulation highlighting the possible implications regarding the accuracy of transport processes associated mainly with ozone trends in the lower stratosphere as seen in recent studies. Apart from the question of which nudging scheme appears to better reproduce MERRA-2 (and to a second degree the free-running WACCM) spatial and temporal upwelling characteristics, the study comprehensively investigates the attribution of these discrepancies by shedding some light on the physical drivers of the upwelling trends. Nudging acts as an additional non-physical tendency in the model equations and it is quite important to evaluate their potential artificial effects on the model dynamics. Therefore, this study greatly improves the understanding regarding the degree of impact arising from the choice (or not) of a particular nudging scheme, albeit in a single model framework. For the reasons above, I recommend this study to be accepted and published with minor revisions. There are a few points that I think they should be addressed by the authors, more to do with enhancing the introduction of the paper by adding a substantial amount of discussion on the nudging studies literature.*

*Specific comments*

*1. I find the introduction to be relatively short and lacking in terms of literature related to nudging studies. In order to set the scene better and highlight that nudging studies are not just used for lower stratospheric ozone trends, more references would be extremely valuable to the reader. One of the first attempts to obtain a comparison between a GCM relaxed towards analyses and the analyses themselves is detailed in Jeuken et al. (1996). There are multiple studies looking at specific meteorological events, such as van Aalst et al. (2004) looking into the Arctic winter transport processes at the end of the 20th century or the SSW during 2009-2010 winter in Akiyoshi et al. (2016). Similarly, nudged simulations were used to focus on the effects of volcanic eruptions on stratospheric tracers such as water vapour in Loffler et al. (2016), to infer the global-mean volcanic effective radiative forcing over the satellite era in Schmidt et al. (2018), as well as to estimate the chemical effects of monsoon circulations on volcanic sulphur*

*particles seen in Solomon et al. (2016). Additionally, Solomon et al. (2015), studied the polar ozone depletion in 2011, using a nudged version of WACCM. In fact, the latter three studies, used a previous version of CESM1-WACCM which is nudged towards an older version of the reanalysis (MERRA), with and without nudging the temperature respectively. A bit of discussion regarding the differences between the nudging schemes used in the aforementioned studies and in the current study certainly wouldn't hurt. I would also recommend some discussion with respect to the differences in between a CTM and an CCM-SD model run as nudging, especially when trying to interpret the differences between Ball et al. (2018) and Chipperfield et al. (2018). CTMs are forced directly with the full 3-D circulation from reanalyses and after many years of optimizations they have been proven quite successful at simulating stratospheric tracers on various timescales (Chipperfield, 1999; Mahieu et al. 2014). On the other hand, CCMs are much more recent tools and exhibit deep-rooted differences when compared to a CTM when looking at their tracer advection, and some discussion regarding their differences would be also a good idea.*

Thank you very much for this reference list! We have included these references, in addition to a few others, in the revised introduction. We tried to keep our introduction short and to the point, but realize it may have left out some details.

We have expanded the introduction on lines 48-57 (a discussion of different "flavors" of nudging), lines 66-67 (errors in nudging schemes), and lines 71-76 (the sensitivity of residual circulation trends and CTM's). We think this will give a better background on the differences between CTMs and nudged CCMs and the difficulties in constraining the residual circulation.

*2. There is a hint in the study that the overarching aim of the study is to capture the free-running WACCM climatology rather than the climatology of MERRA2 - some justification is required if that is the case. Although the research question (reproducing the residual circulation variability and trends of MERRA2) of the study is clearly stated in the introduction (lines 56-57), there are various places across the text where the message appears to be the reproduction of free-running WACCM. As an example, it is stated that the nudging will create conflicts due to the differences between WACCM and MERRA2 underlying climatologies (explained in section 2 - lines 88-92). I would suggest rephrasing the relevant parts where the point seems to be lost. The point of nudging is exactly to reproduce the reanalyses themselves albeit exhibiting spurious features in*

*the stratospheric residual circulation. It should be noted that between reanalysis products and among different estimates of wbar\* there lies significant uncertainty with respect to upwelling trends as seen in Abalos et al. (2015) as well as Kobayashi and Iwasaki (2016).*

As we discussed in our response to Referee #1, we placed the goal before the motivation in our discussion of (not) nudging the mean. Rather than suggesting that preserving the climatology is the goal, we know explicitly state our hypothesis on lines 78-81 that any differences in the climatology between the input meteorology and the model may lead to spurious circulations and trends.

Thank you for suggesting we discuss the variability of the residual circulation across reanalysis products; lines 74-76 connect the variability in the residual circulation trends to some recent work suggesting the tropospheric meridional circulation in reanalyses is in an unphysical balance.

*3. There's no mention of the MERRA2 output the authors have used throughout this study. How did you calculate the TEM diagnostics? Did you perform the calculation on the native MERRA2 levels? What output have you used? How about its temporal frequency? This information needs to be included by describing all the above in either section 2 or section 3.*

Thanks, this was an oversight. We now state on line 97 that we use the ASM product, and on lines 198-199 state that eddy fluxes are calculated from 3-hourly instantaneous output on native levels.

**Minor comments**

*Line 22 : See discussion above (point 2)*

Here and elsewhere, we've cleaned up the manuscript to better reflect our hypothesis.

*Line 55 : Some more references are needed here such as Abalos et al. (2015) and Kobayashi and Iwasaki, (2016) with findings related to the discrepancies in the trends of the residual circulation between reanalysis.*

Thanks - these references better connect the Chemke and Polvani paper to the residual circulation and its trends.

*Line 62 : In the context of this study is quite clear that nudging zonal-mean temperatures alters the meridional eddy momentum and heat fluxes in the TTL without being successful in simulating the underlying MERRA2 trends. Applying a thermal nudging (temperature) could potentially lead to a sustained spurious heat source in the model, which leads to a stronger BDC in the lower stratosphere as seen in Miyazaki et al. (2005) with a different model. However, the last sentence is quite strong as a statement and generalizes a result which is model specific. Therefore, I would recommend rephrasing this bit so it doesn't strike as misleading.*

We agree this was too strong of a statement and now say, "We find that not nudging zonal mean temperatures results in the best reproduction of upwelling trends, while nudging zonal mean temperatures tends to degrade these trends, consistent with our hypothesis."

We have added Miyazaki et al. [2005] to our discussion of temperature trends and spurious heating on line 519.

*Line 69 : When nudging, the choice of the relaxation timescale can play an important role (Merryfield et al. 2013), although there is no consensus that a specific timescale necessarily leads to an improvement (Hardiman et al. 2017). I'm aware of a standard full WACCM-SD CCMI simulation using a nudging timescale of 5 hours - towards MERRA though (Orbe et al. 2018; Orbe et al. 2019 under review in ACPD), have you performed any additional runs with this timescale (or have plans) to compare with?*

We did attempt to run with a more aggressive nudging timescale of 5 hours (10% per timestep). However, this required a substantial increase in the physics timestep subcycling to prevent instabilities in the convection scheme. The resulting simulations would not be numerically comparable to our existing simulations, so we did not investigate the issue of nudging timescale. This is now discussed on lines 92-96.

*Lines 84-87 : It would be very helpful (at least to me) if you could include a figure (in the supplement) showing the vertical profiles of the pressure levels in WACCM-L66/L88 and MERRA2 to better highlight their spacing differences throughout the depth of the atmosphere.*

That's a great idea - we've added two versions of this figure, one with a linear pressure scale and one with a logarithmic pressure scale to emphasize the middle and upper atmosphere.

*Lines 98-99 : Here you mention that 6-hourly MERRA2 anomalies are used for nudging WACCM, I assume you are interpolating in time to nudge every 3 hours (see line 71)?*

This was an error, we used 3-hourly output in all cases. The model then interpolates the 3-hourly output to the current model time to determine the nudging terms.

*Line 105 : Please clarify that this improvement refers specifically to aerosol-climate interactions in Zhang et al. (2014).*

We have rewritten all references to preserving the mean climate of WACCM, so that it is clear that we see it as a potential solution to the problem of reproducing TTL trends - not something that is an end in and of itself. Here, we are simply citing Zhang et al. [2014] to give credit to the idea of climatological anomaly nudging - hopefully there is no sense of an implied "improvement" that needs clarification.

*Lines 155 & 159 : Please correct the reference - it is : Hardiman et al. (2010)*

Thanks, this was in error.

*Line 164 : SF6 can be considered linear only by approximation, characterised by a fast growth rate and there needs to be a correction for this. See Garcia et al. (2011).*

Thanks, we've added this reference and a clarification to line 217.

*Line 224 : This line refers to Figure 3b, where you calculate the difference in the trends compared to MERRA2. For clarity, it would be better to rephrase and not use the word negative but either smaller or bigger to reflect their differences. E.g. in the TTL all WACCM runs + MERRA 2 have positive trends as seen in Figure 3a, and the term negative might be misleading as the trends are just smaller (but still positive).*

That's a good point, it was somewhat ambiguous the way it was rewritten. It now states "The upwelling trends in all WACCM runs tend to be smaller in the TTL and larger aloft" on line 285.

*Line 228 : " The standard UVT…" - Clarify that this holds true for both versions (L66 + L88).*

Yes, good idea, it now states "The UVT L88, UVT, and UVT climatological anomaly…".

*Lines 241 - 252 : Excellent discussion (and figure)!*

Thank you.

*Line 309 : By gravity wave (GW) momentum forcing are you referring to all the parameterizations? Meaning orographic (OGW) + non-orographic (NOGW) gravity wave drag put together? Please clarify.*

Yes, correct, it's the sum of all gravity wave sources in the model. We have clarified this by stating, "However, gravity wave momentum forcing from orographic and non-orographic waves…"

*Line 311 : McLandress and Shepherd (2009), using CMAM, show the total contribution of both resolved and parameterized wave drag occuring at the edge of the pipe in the lower stratosphere in boreal winter in their figure 18. However, this lumps together resolved (major contributor at the edge of the pipe) and all GW parameterizations while the orographic gravity wave drag contributes more in the NH mid-latitudes instead. I would suggest caution drawing parallels to this result which remains model specific. Different parameterizations lead to various magnitudes of contributions to the upwelling throughout the stratosphere and specifically for the versions of CMAM over the past decade, it has been shown that the NOGW contributes negatively in the upwelling in the lower stratosphere in SPARC, 2010 and more recently in Chrysanthou et al. (2019).*

This is a fair point - it does appear this is a case where different models and different generations of models provide different answers. We aren't focused on the physical plausibility of any of the trends here, merely their correspondence with those in MERRA2, so rather than expand the discussion to note this is not a robust feature of models, we have simply eliminated it.

---

## Author Comment (AC3) · 21 Dec 2019

**Responses to Referee #3**

*Many studies have been performed whereby otherwise freely-running chemistry-climate models have had the day-to-day evolution of the dynamical fields constrained to follow the historical evolution as represented by reanalysis datasets. Here, Davis et al. present an analysis of different nudging schemes, using different combinations of variables or only nudging to zonal anomalies that are calculated in different ways, to assess the impact on the residual circulation of the lower stratosphere with a particular emphasis on how the nudged simulations differ with freely-running simulations and with the reanalysis dataset used for nudging. The study is very nicely performed and includes a convincing mechanistic diagnosis of the ways in which nudging of different variables affects the trends in tropical upwelling.*

*I really have no major concerns on the methodology or analysis presented here and my comments are mostly minor. One concern I do have, however, is the presentation of the effects of nudging zonal mean temperature on reproducing trends. In the abstract, at lines 21 – 23, the authors state that nudging to anomalies better reproduces trends in stratospheric upwelling, period. Taking a broader view, it would seem that nudging anomalies produces trends in upwelling that are more similar to the trends produced by the free-running (AMIP) simulation. This is clearly shown in Figure 3a, where the schemes that involve nudging to anomalies are much closer to the free-running simulation both in the TTL and in the lower stratosphere. As a consequence of the trends produced by the free-running AMIP simulation, the simulations nudged to zonal anomalies agree better with MERRA2 in the TTL but agree more poorly through the lower stratosphere. The degree of differences to the AMIP simulation across different nudging schemes also extends to the analysis of the EP-flux trends where it is stated (lines 392 - 394) that 'the response of the "no zonal-mean temperature nudging"simulations can be understood as the superposition of the "zonal-mean temperature nudging" simulation response - a slightly-incorrect MERRA2 response – and the AMIP response.' From both the analysis of trends and the analysis of the mechanism it appears that the response of schemes that do not affect the zonal-mean temperature produce trends that are more like the AMIP free-running simulation. I would suggest the authors should not overstate the conclusions of the effects of nudging temperature on the ability of the nudged model to reproduce the trends in the reanalysis as it would seem to depend significantly on the underlying behaviour of the free-running simulation.*

Thanks - this is a good point. It was not consistent to discuss the EP flux results as AMIP-like or MERRA2-like but not discuss upwelling trends in this way.

In the abstract, we now state "None of the schemes substantially alter the structure of upwelling trends - instead, they make the trends more or less AMIP-like."

The revised discussion of Figures 3 (lines 303-304) makes clear that the zonal anomaly nudging is only superior in the TTL.

*Not at all a criticism, but more of a puzzled commentary. Figure 11 shows that nudging the zonal mean of temperature from MERRA produces temperature trends that disagree with the trends in MERRA. I can accept that the cause of the differences in the stratophere are not fully understood and may be related to unintended secondary circulations, but the anomalous trends found only in simulations that nudged the zonal mean temperature extend deep into the troposphere. In fact, the trends in the upper tropical troposphere appear to be three or four times larger than the trends in the same region found in MERRA2. Do you have any explanation for the discrepancy in trends in the troposphere and could there be links to the trends in the lower stratosphere?*

The temperature difference between AMIP and MERRA2 maximizes around the level of net zero radiative heating in the TTL, where longwave cooling is close to zero and shortwave heating is at a minimum, whereas above and below the TTL the longwave cooling is substantially stronger [Fueglistaler et al. 2009]. So because the radiative terms are so small, it may be that this region is particularly sensitive to temperature perturbations and can more rapidly convert temperature nudging to perturbed heating. How this drives trends is quite unclear to us.

The extension into the troposphere may have something to do with convective parameterizations. MERRA2's temperatures (and any meteorological input data set's temperatures) will have convective effects baked in, so that nudging WACCM to those temperatures will result in a kind of double-counting as WACCM also has convection. Again, how this could contribute to the "wrong" temperature trends is unclear, but it could certainly present an inconsistency. It's also possible that the incorrect trends in the TTL are just the decaying signal of this problem in the troposphere.

We think future work using a single model, like Smith et al. [2017], would be a more self-consistent system and might be the best avenue for understanding this problem.

***Minor comments***

*Lines 98 – 99, for the case where WACCM is nudged towards anomalies it is stated 'To generate the nudging input, 6-hourly MERRA2 U, V, and T anomalies are calculated...'but a bit earlier, at lines 68 – 71, when the default nudging scheme is described it is stated that the MERRA2 reanalysis is supplied to the model every 3 hours. Is this difference real or just a typo? And if it is real, have the authors considered the differences in model behaviour that may be caused by reducing the frequency by a factor of two? Part of the motivation behind pointing this out is an open question about the effect of linearly interpolating in time between the available reanalysis.*

This was indeed a typo and has been fixed - we use the 3-hourly output in all cases. However, the model's nudging scheme does interpolate the meteorological input to the current model time (lines 98-99).

*Line 180 – Figure 1, I might suggest reducing the vertical extend to maybe 5 hPa so that the horizontal scale can be expanded. None of the other graphs extend beyond 30hPa.*

While it's true the vertical extent is substantially higher than in the other plots, our intent was to begin with a macroscopic view of the whole stratosphere and display the large-scale structure of upwelling (the mass flux monotonically decreasing with height throughout the stratosphere), the remarkably rapid decrease with height of upwelling through the TTL compared to all heights above (indicating the strong poleward flow in the shallow branch), and the consistency of WACCM vs. MERRA2 upwelling (e.g., AMIP essentially always has more upwelling, throughout the entire stratosphere). We feel that including the log-scale difference plot alleviates the need to expand the axis, as it emphasizes the differences lower in the stratosphere.

*Line 216 – missing 'A' in 'MERR2'*

Thanks, this has been fixed.

*Lines 239-240: Here it is stated that 'This all demonstrates that (incidentally) nudging zonal-mean MERRA2 temperatures has a negative impact on the upwelling trend morphology and magnitude.' I see how the findings of the correlation coefficient of trends with MERRA2 being largest for simulations that do not nudge the zonal-mean temperature supports the statement on morphology. But the magnitude of the trend*

*over large regions of the vertical profile shown in Figure 3 is closest to MERRA2 for the simulations that do nudge zonal temperature. The magnitude of the trends in UVT is closer to MERRA2 than UV, and UVT(ca) is closer than UV(ca) between 90 hPa and 40 hPa. While the magnitude of the trends in UVT(za) are the furthest from MERRA2 everywhere above 90 hPa. The experiments where zonal average temperature is not nudged are closer to the AMIP simulation and this is an advantage in the TTL as the AMIP simulation has the largest positive trends and is thus closest to MERRA2. But producing trends closer to the freely-running AMIP simulation becomes a disadvantage higher up where the freely-running AMIP simulation produces more positive trends than MERRA2. Having read a bit further, I see how you eventually address this (and I particularly like Figure 4) but the statement at Lines 239-240 about the effect of nudging zonal-mean temperature on the magnitude of trends seems unsupported.*

Thanks, another reviewer made this point as well. We have edited the discussion to be more specific to the TTL, and to make the point that the recent ozone trends in Ball et al. [2018], part of the motivation of this work, depend on the dynamics in this region being accurately resolved.

"This all demonstrates that (incidentally) nudging zonal-mean MERRA2 temperatures - the UVT, UVT L66, and UcaVcaTca simulations - has a negative impact on upwelling trend morphology and magnitude in the TTL. While it is true that the trends in the zonal anomaly nudging simulations are too positive above the TTL, key for constituent transport into the stratosphere and for recent ozone trends is the upwelling trend at and above the tropopause."

*Lines 284 - 285: It is stated here that the poor performance of the zonal anomaly nudging in reproducing variability in upwelling below 85 hPa suggests 'a strong role for the zonal-mean circulation in transforming wave dynamics into zonal-mean momentum forcing and therefore upwelling (Fig. 6).' Are you suggesting that the QBO has a role to play in upwelling in the TTL? Is there anything to be seen correlating the MERRA2 variability with that of the AMIPQBO run? [ Okay, way down at Line 460 I see where you address the role of the QBO on variability in the TTL using UVT(za) nudging. ]*

Right, sorry that we leave this idea until the end of the paper. We tried to be linear in our discussion of the results, but obviously there are many cases like this where we don't revisit an idea until later.

*Line 363 – 366 – The caption for Figure 9 does not mention what is indicated by the thick black line. Is it the lapse rate tropopause?*

Thanks - that is correct, and we have added this to the figure caption.

*Line 414 – minor typo on '\hypothesize'*

Thanks, fixed.

---

## Author Comment (AC6) · 21 Dec 2019

[Figure]

*Figure S1: Vertical levels in MERRA2, WACCM on MERRA2 levels with 16 free-running levels above MERRA2's lid, and WACCM on native levels, assuming a 1000 hPa surface pressure.*

[Figure]

*Figure S2: As in Fig. S1, but on a logarithmic pressure grid.*

[Figure]

*Figure S3: Climatological (left) gravity wave drag and (right) latent heating averaged between 25S and 25N for selected runs. Behavior of UV nudging simulations is quantitatively similar to their UVT counterparts. Gravity wave drag is the sum of the output variables UTGWSPEC, BUTGWSPEC, and UTGWORO.*

[Figure]

*Figure S4: Frequency of upwelling based on the streamfunction definition of the residual circulation for each model simulation for December-January-February only. Hatching indicates there is upwelling 100% of the time.*

[Figure]

*Figure S5: As in Fig. S4, but for June-July-August only.*